**Perspective**

# The biogeochemical transport by the Gulf Stream

Richard G. Williams [1] ✉, Peter J. Brown [2], Yohei Takano[3], Gaël Forget [4], Dani Jones [3,5], Anna Katavouta [6], Elaine McDonagh [2,7] & Vassil M. Roussenov[1]

The Gulf Stream is important for the climate system through its transport and air-sea exchange of heat. What is less well accepted is the role of the Gulf Stream in the carbon cycle. Here we examine how the Gulf Stream provides a "biogeochemical stream", a sub-surface horizontal flux carrying waters with high concentrations of nutrients and low concentrations of anthropogenic carbon. Model experiments reveal particles released in dense layers at the start of the Gulf Stream follow trajectories extending into the subpolar gyre, while particles released at the surface are confined to the subtropics. Following a pathway to the subpolar gyre, the biogeochemical stream carries older, nutrient-rich and anthropogenically carbon-depleted waters along density layers and, when those dense layers outcrop into the mixed layer, enhances the subpolar drawdown of atmospheric carbon. This connectivity is supported by model sensitivity experiments revealing the subpolar upper ocean carbon content and upstream dense waters in the Gulf Stream connecting on timescales of 4 to 8 years. The likely effect of climate change on the biogeochemical stream is a decrease in the delivery of these older waters, both high in concentrations of nutrients and depleted in anthropogenic carbon, to the subpolar mixed layer, so weakening future North Atlantic carbon uptake from the atmosphere.

Climate projections suggest that natural carbon sinks involving the global ocean and land are expected to diminish in their effectiveness in curbing the rise of atmospheric $CO_2$[1,2]. The global ocean response to carbon emissions is widely viewed as being a consequence of surface warming leading to reduced solubility and increases in local stratification[3] that reduce ventilation and vertical nutrient supply. However, this viewpoint ignores the crucial role of western boundary currents in redistributing older waters around ocean basins with elevated concentrations of nutrients and depleted concentrations of anthropogenic carbon.

The North Atlantic is one of the most effective locations in the global ocean for carbon uptake from the atmosphere and long-term carbon storage: north of 25°N, the region accounts for 23% of global air-sea $CO_2$ fluxes and 15% of the global anthropogenic carbon inventory, despite covering only 7% of the surface area[4–6]. This uptake of atmospheric $CO_2$ is connected to extensive surface heat loss[7] and biological carbon drawdown over the North Atlantic (Fig. 1). The surface heat loss is a consequence of the northward delivery of warm, surface waters, while the biological production is a consequence of the northward supply of nutrients[8], so that both are affected by the Atlantic Meridional Overturning Circulation. Air-sea $CO_2$

fluxes over the North Atlantic are then potentially susceptible to impacts of circulation change and Arctic freshwater discharge.

The Gulf Stream is a major part of the upper limb of the Atlantic Meridional Overturning Circulation. While the role of the Gulf Stream in the northward transport of heat is widely recognised, its contribution to mitigating human-driven atmospheric carbon increases is not. The Gulf Stream provides a sub-surface redistribution of nutrients and waters depleted in anthropogenic carbon over the North Atlantic, which is here referred to as a "biogeochemical stream". This biogeochemical role of the Gulf Stream is relevant to the wider issue of how the ocean carbon system responds to climate change[9].

The role of the Gulf Stream for heat and nutrient transfer is next reviewed, followed by an assessment of the effect of the Gulf Stream on the carbon cycle, by considering the biogeochemical properties in the Florida Straits, their downstream trajectories and property evolution over the North Atlantic. The sensitivity of the subpolar carbon content to upstream transport of carbon is then examined through new adjoint model experiments. Finally, future possible changes in carbon uptake from the atmosphere for the North Atlantic are separated into competing contributions from the rising

[1]Department of Earth, Ocean and Ecological Sciences, School of Environmental Sciences, University of Liverpool, Liverpool, UK. [2]National Oceanography Centre, Southampton, UK. [3]British Antarctic Survey, Cambridge, UK. [4]Massachusetts Institute of Technology, Cambridge, MA, USA. [5]Cooperative Institute for Great Lakes Research, University of Michigan, Ann Arbor, MI, USA. [6]National Oceanography Centre, Liverpool, UK. [7]NORCE, Bergen, Norway. ✉e-mail: ric@liverpool.ac.uk

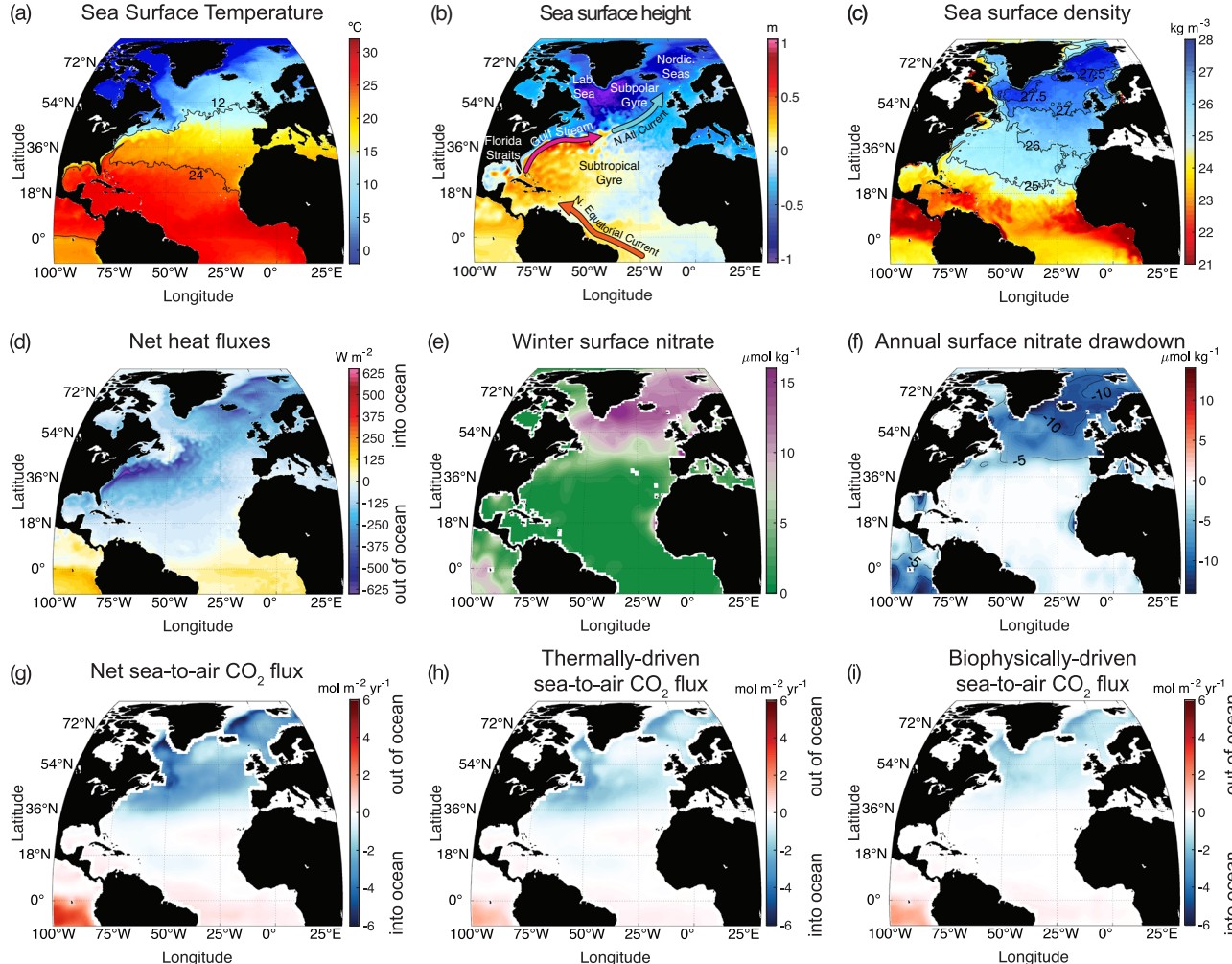

**Fig. 1 | Gulf Stream environment.** Surface fields for the North Atlantic: (**a**) sea surface temperature (°C), (**b**) dynamic height (m) and (**c**) sea surface density - 1000 (kg m⁻³), for December 2022 (from Operational Mercator global ocean analysis and forecast system[53] at 1/12° horizontal resolution) with in (**b**) red arrows providing a schematic view of western boundary flows and blue arrow the North Atlantic Current; (**d**) net air to sea heat flux (negative is ocean heat loss) for December 2022 at 1/4° resolution[54] (W m⁻²); (**e**) winter surface nitrate[55] ($\mu$mol kg⁻¹); (**f**) annual surface nitrate drawdown (from winter to summer contrast)[72] ($\mu$mol kg⁻¹); (**g**) annual sea-to-air $CO_2$ flux at 1° resolution (mol m⁻²yr⁻¹) (positive is out of the ocean, negative is into the ocean) for 2019[56], which is separated into (**h**) thermal and (**i**) biophysical components.

atmospheric $CO_2$ and climate change (encapsulating changes in temperature, stratification and circulation). Our viewpoint of biogeochemical streams affecting the ocean carbon sink is relevant where there are similar western boundary currents and separated jets in the rest of the global ocean.

## Gulf Stream and the climate and carbon system
### Gulf Stream and heat transfer
The Gulf Stream is widely viewed as being important in the climate system through its release of heat to the atmosphere[10] and transport of heat within the ocean[11–14] (Fig. 1a, d). The Gulf Stream advects warm tropical waters along the western boundary of the North Atlantic, separates from the coast at Cape Hatteras and its extension transports warm waters into the rest of the basin; the Gulf Stream extension after passing the Grand Banks is called the North Atlantic Current[15] (Fig. 1b).

Over the Gulf Stream extension, there is a maximum in the net heat loss to the atmosphere (Fig. 1d) and the strong sea surface temperature gradient across the fast-flowing current helps localise atmospheric storm tracks[16,17]. The transport of the Gulf Stream is made up of the return flow of the wind-driven subtropical gyre circulation, augmented by tight local recirculations[18], plus the northward upper flow of the meridional overturning circulation[13,19]. The Gulf Stream makes a crucial contribution to

ocean heat transport by being part of the upper limb of the meridional overturning circulation, which carries warm water northward and colder deep water southward[20]. This heat transfer is important in warming the atmosphere with typically half of the ocean heat carried by the meridional overturning circulation at 25°N being taken up by the atmosphere by 50°N[14,21]. This ocean transport and release of heat to the atmosphere[22], augments the seasonal heat release from the ocean to the atmosphere[23], and together contributes to the mild winter climate of western Europe.

While the effect of the Gulf Stream on the climate system has had extensive investigation, there has been less attention on its effect on biogeochemical cycles, such as on basin-scale patterns in annual nutrient utilisation (Fig. 1f) and annual air-sea $CO_2$ exchange over the North Atlantic (Fig. 1g), which includes both thermal and biological contributions (Fig. 1h, i). We next address the role of the Gulf Stream in transporting nutrients and then the redistribution of carbon; these effects together may be viewed in terms of a biogeochemical stream.

### Gulf Stream and downstream nutrient transfer
The Gulf Stream is recognised as providing a nutrient transport over the basin, referred to as a "nutrient stream[24,25]. To explain this phenomenon,

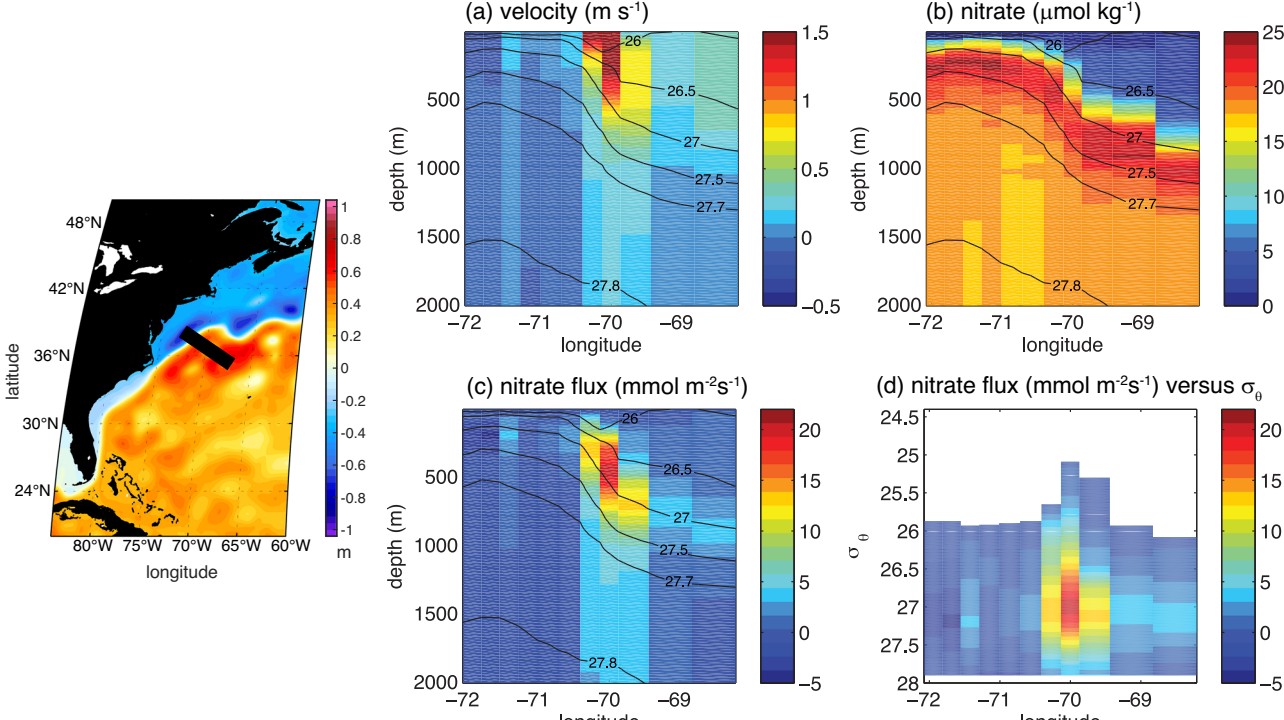

**Fig. 2 | Nutrient Stream from observations of the separated Gulf Stream.**
Observations of the nitrate transport by the Gulf Stream[26] across a section at 36.5°N
in May 2005. **a** geostrophic velocity (m s$^{-1}$); (**b**) nitrate ($\mu$mol kg$^{-1}$); (**c**) and (**d**)
Horizontal nitrate flux along the Gulf Stream (mmol s$^{-1}$m$^{-2}$) versus depth (m) and

potential sigma, $\sigma_\theta$ (kg m$^{-3}$), evaluated from the product of (**a**) and (**b**). Potential
density contours included from 26 to 27.8 kg m$^{-3}$ in (**a**) to (**c**). The inset shows the
dynamic height[53] (m) together with the section (black line).

consider observations along 36.5°N, where the separated Gulf Stream
is characterised by having fast surface velocities reaching greater than
1.5 m s$^{-1}$ (Fig. 2a).

While there is a clear sea surface temperature signal across the Gulf
Stream, there is little contrast in surface nutrient concentrations (Fig. 1e).
The nutrient stream of the Gulf Stream is defined by the horizontal flux of
nutrients, from the product of the horizontal velocity and nutrient con-
centration (Fig. 2a, b), and has a subsurface maximum[26] at depths of between
300 m and 700 m and in a potential density range from $\sigma_\theta$= 26.5 to 27.5 kg
m$^{-3}$ (Fig. 2c, d). The nutrient stream provides a nutrient communication
pathway over the basin[24,25,27].

The downsteam pathway of the nutrient stream is evident in an eddy-
permitting model solution, revealing maxima in the horizontal nitrate and
dissolved organic nitrogen fluxes extending along the Gulf Stream within
the subtropical gyre for light density layers (Fig. 3a, b) and extending into the
subpolar gyre for dense layers[26] (Fig. 3c).

The nutrients carried in the nutrient stream consist of preformed
nutrients (defined as being supplied from the mixed layer) originating
mainly from outside the subtropics[28] and regenerated nutrients supplied
from biological fallout and regeneration; the preformed nutrients them-
selves are primarily supplied from mode and intermediate waters[29] origi-
nating from the Southern Ocean (Fig. S1d).

The nutrients in the nutrient stream are transported within density
layers that eventually outcrop at the surface, either with light layers out-
cropping over the northern flank of the subtropical gyre or denser layers
outcropping in the subpolar gyre[26,30] (Fig. 3d). This outcropping then
transfers the nutrients into the winter mixed layer[29], ultimately helping to
sustain high-latitude biological productivity (Fig. 1e, f).

In summary, the nutrient stream[24] provides a mechanism to supply
nutrients to the winter mixed layer[29] of the subpolar gyre and so helps sustain
biological productivity there[25,31]. This nutrient transport mechanism is con-
sistent with a tracer-based view that nutrient supply from the Southern
Ocean[32] ultimately sustains the biological productivity of the northern basins.

## Gulf Stream and downstream carbon transfer
Drawing upon the role of the Gulf Stream in transporting nutrients, we next
consider the effect of the Gulf Stream on the redistribution of carbon. Start
by considering observations in the Florida Strait at 27 °N on the western
boundary of the North Atlantic. The Florida Current, forming the start of
the Gulf Stream by the coast, carries high concentrations of nutrients in sub-
surface fresher, denser waters (with neutral density, $\gamma > 26.8$ kg m$^{-3}$)
(Fig. 4a–d); typically half of the nutrient concentrations are preformed and
half regenerated from biological fallout (Fig. 4e). The combination of the
velocity and nutrient concentrations leads to high northward fluxes of
nutrients (Fig. 4f).

The Gulf Stream also carries high concentrations of dissolved inorganic
carbon (DIC) in fresher, denser waters and high concentrations of
anthropogenic carbon in lighter waters, which are transported northward
(Fig. 4g-i). The fresher, denser waters are older and carry less anthropogenic
carbon (having been last been in contact with the atmosphere when
atmospheric $CO_2$ was much lower). Hence, there is a capacity to hold
additional anthropogenic carbon when these density layers outcrop in the
mixed layer and are next in contact with the atmosphere (Fig. 4j, k).

The strong northward flow in the Florida Current then provides
subsurface maxima in the northward fluxes of nitrate and the capacity to
hold additional anthropogenic carbon (Fig. 4f, l). These subsurface maxima
in the horizontal property fluxes define the biogeochemical stream.

The importance of this biogeochemical stream to the rest of the North
Atlantic depends on the downstream circulation pathways.

## Connectivity of the Gulf Stream and the subpolar gyre
There is an expectation that properties carried in the Gulf Stream directly
connect with the rest of the North Atlantic. However, there is a surprising
conundrum that almost no surface drifters released in the subtropical gyre
are observed to pass into the subpolar gyre[33]. Lagrangian particle, numerical
model experiments reveal that this lack of surface exchange between the
gyres is due to the surface wind-driven, Ekman transport being directed

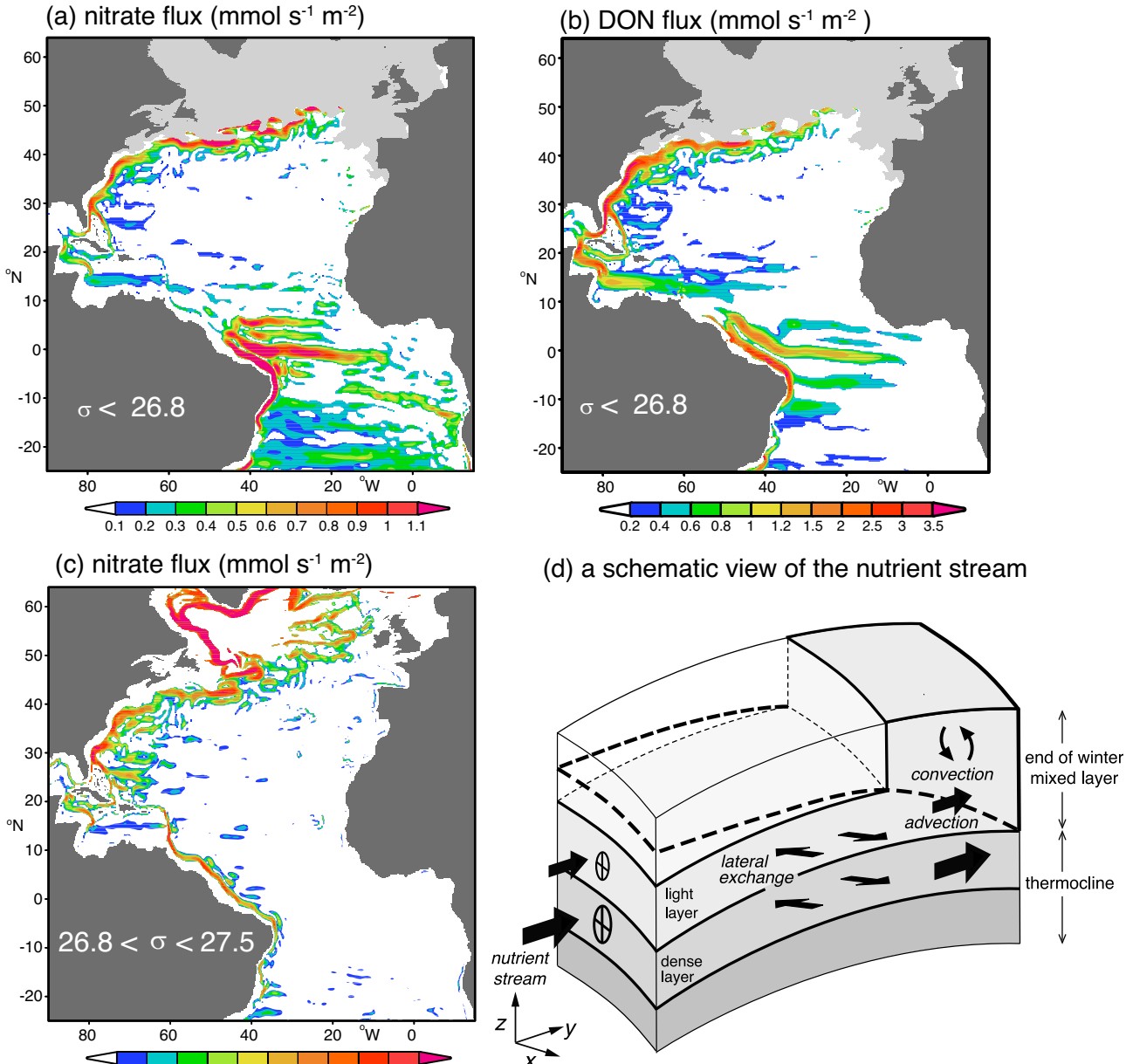

**Fig. 3 | Nutrient Stream over the Atlantic basin from an eddy-permitting model[26].**
**a** horizontal nitrate flux (mmol s$^{-1}$m$^{-2}$) integrated over density layers less than
$\sigma = 26.8$ kg m$^{-3}$; (**b**) horizontal dissolved organic nitrogen (DON) flux (mmol
s$^{-1}$m$^{-2}$) for the same layers as in (**a**); and (**c**) horizontal nitrate flux (mmol s$^{-1}$m$^{-2}$) for
density layers integrated over $26.8 < \sigma < 27.5$ kg m$^{-3}$. The modelled horizontal
nitrogen flux is from an isopycnic model (MICOM) with 0.23$^o$ horizontal resolution
coupled with a biogeochemical model. The associated meridional transports across
the basin and a global schematic are shown in Fig. S1. In (**d**), a schematic figure of a
nutrient stream (black arrow) transferring nutrients within a series of density layers
(shaded) lying within the thermocline into the downstream mixed layer at the end of
winter (base denoted by thick dashed line), where convection redistributes the
nutrients in the vertical[73]. The nutrient stream within a light layer remains confined
within the subtropical gyre, while the nutrient stream in denser layers passes into the
subpolar gyre and provides a nutrient input into the subpolar mixed layer.

southward at the inter-gyre boundary and that inhibits a northward surface
exchange[34]. Likewise, there is a limited exchange of sea surface temperature
anomalies across the inter-gyre boundary[35]. There is though communica-
tion of particles and heat within subsurface density layers from the sub-
tropics to the subpolar latitudes, since those layers are not impacted by the
surface Ekman transport[34,35].

To illustrate how the biogeochemical stream operates, we conduct
particle-tracking experiments releasing 10,000 virtual particles in the Florida
Strait using an observationally-optimised ocean circulation model with the
particles advected by the three-dimensional circulation (Methods[36]).

Particles released in light waters in the Florida Straits (between the
surface and a depth of 110 m) primarily remain within the subtropical gyre

(Fig. 5a), consistent with observations of limited gyre exchange of surface
drifters[33]. In contrast, particles released in denser waters in the Florida Straits
(at depths from 110 m to 380 m) are almost evenly split between those
staying in the subtropical gyre and those passing into the subpolar gyre,
while particles released in the most dense waters (at depths of 380 m to 860
m) nearly all pass into the subpolar gyre (Fig. 5b, c). The denser particles
shoal as they move from the subtropics to the subpolar gyre (Fig. 5c,
changing from yellow to orange) and then deepen around the northwest
flank of the subpolar gyre (Fig. 5c, changing from orange to blue) or pass
futher north into the Norwegian Sea. Hence, deeper pathways connect water
masses and biogeochemical properties in the Florida Current to subpolar
latitudes of the North Atlantic.

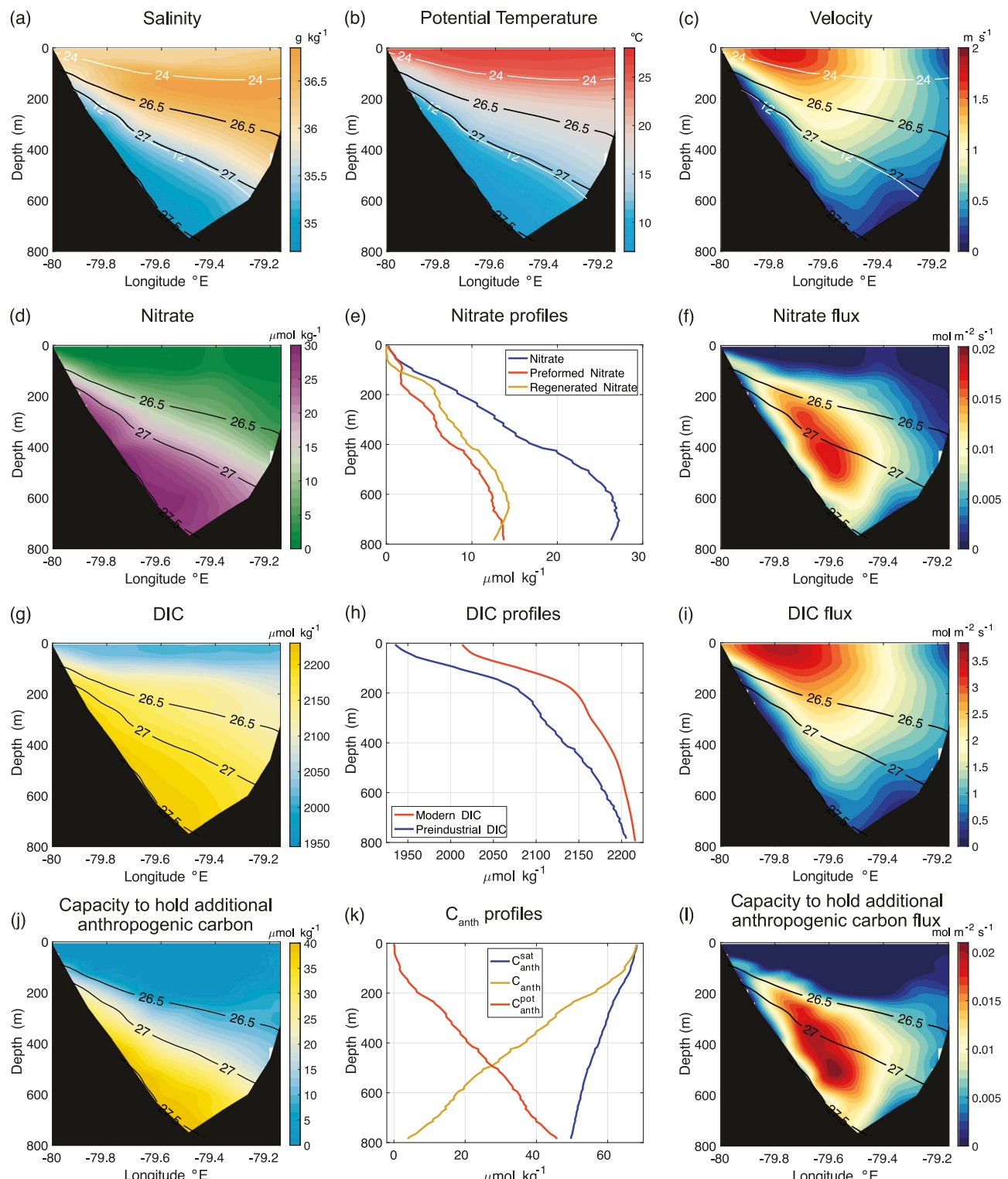

**Fig. 4 | Biogeochemical Stream in the Florida Straits at 27ºN. a** salinity (psu) (with white contours for 12ºC and 24ºC); (**b**) potential temperature (ºC); (**c**) meridional velocity (m s$^{-1}$); (**d**) nitrate concentration ($\mu$mol kg$^{-1}$); (**e**) nitrate profile (blue) with preformed (red) and regenerated (yellow) components ($\mu$mol kg$^{-1}$); (**f**) northward nitrate flux (mol m$^{-2}$s$^{-1}$); (**g**) dissolved inorganic carbon (DIC) ($\mu$mol kg$^{-1}$); (**h**) profiles of modern DIC (red) and preindustrial DIC (blue) ($\mu$mol kg$^{-1}$), where modern DIC equals the sum of the preindustrial DIC and the anthropogenic carbon; (**i**) northward DIC flux (mol m$^{-2}$s$^{-1}$); (**j**) capacity of waters to hold additional anthropogenic carbon (anthropogenic carbon uptake potential, $\mu$mol kg$^{-1}$); (**k**) anthropogenic carbon profile (yellow) and the maximum anthropogenic carbon that could be held if the waters are saturated (blue), and their mismatch defines the capacity to hold additional anthropogenic carbon (red); and (**l**) northward flux of capacity to hold additional anthropogenic carbon (mol m$^{-2}$s$^{-1}$). The plots include neutral density contours 26.5 to 27.5 kg m$^{-3}$.

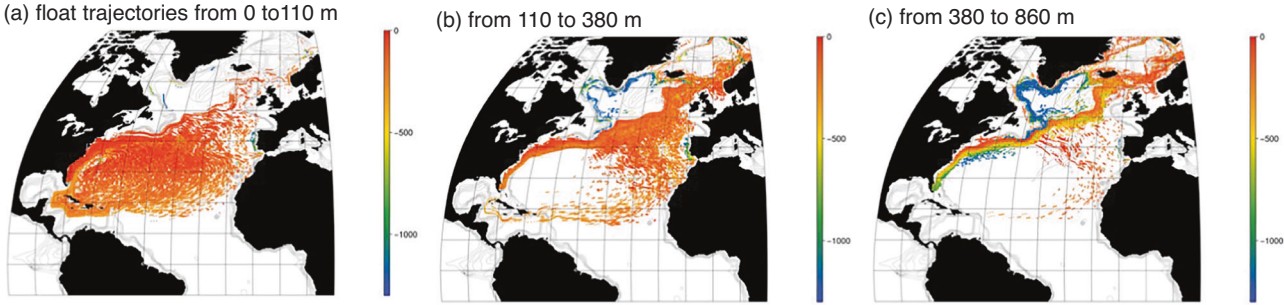

**Fig. 5 | Pathways emanating from the Gulf Stream.** Tracking of virtual seawater particles over a 10-year period. 10000 virtual seawater particles are initially released at Florida Strait in three different depth ranges: (**a**) 0 to 110 m, (**b**) 110 to 380 m, and (**c**) 380 to 860 m. Calculations use the monthly climatological mean three-dimensional flow fields from the ECCO4 release 2 ocean state estimate[36,62]. Colour scale is for the depth of the virtual seawater particles.

### Carbon evolution along a biogeochemical stream pathway

A biogeochemical stream is proposed following a sub-surface pathway connecting the subtropics to the subpolar latitudes (consistent with the particle displacements in Fig. 5c). Our viewpoint is that biogeochemical properties in the Florida Strait (Fig. 4) are carried along this sub-surface pathway and ultimately affect the subpolar uptake of atmospheric $CO_2$.

Along this pathway, sub-surface waters with high concentrations of DIC and nitrate spread northward from the subtropics to the subpolar latitudes along neutral densities 27.0 to 27.5 kg m$^{-3}$ (Fig. 6a, b). During this spreading phase, the DIC decreases in concentration downstream through the effect of physical processes, probably involving diapycnal mixing with lighter waters or mesoscale eddy stirring along density surfaces with lower concentrations of DIC (Fig. 6d).

The air-sea flux of $CO_2$ is locally defined by the difference in the partial pressure of the dissolved $CO_2$ in the surface mixed layer and the atmospheric partial pressure, pCO$_2$, represented by $\Delta$pCO$_2$ (Methods). The dissolved $CO_2$ signal in the mixed layer is affected by the advection of DIC and accompanying temperature and nutrients along density layers and eventual outcrop of these waters (Fig. 6c and e, red bars). Over the subpolar latitudes, the $\Delta$pCO$_2$ signal is positive along the neutral densities 27.0 to 27.5 kg m$^{-3}$ and represents waters that are just over saturated for year 2002 (Fig. 6e, red bars versus grey dashed line). Hence, by itself, this injection of DIC along the density surface into the mixed layer elevates surface pCO$_2$ values, and suggests subpolar $CO_2$ outgassing for year 2002. However, there are additional thermal and biological mechanisms that subsequently affect the surface pCO$_2$ values: a combination of surface heat loss and biological utilisation of nutrients (predominantly preformed) (Figs. 1d–f and 6b) lead to a lowering of surface pCO$_2$ values, generating a negative $\Delta$pCO$_2$. A net annual subpolar ocean uptake of $CO_2$ from the atmosphere from both thermal and biophysical drivers is thus derived[37,38] (Fig. 1g–i).

The total carbon content of these waters also shapes the magnitude of the uptake signal. As the biogeochemical stream waters have been separated from the atmosphere over long timescales, these waters are undersaturated in anthropogenic carbon with respect to a modern atmosphere. Sub-surface waters change from having low concentrations of anthropogenic carbon in the subtropics to progressively higher concentrations in the subpolar gyre along neutral densities 27.0 to 27.5 kg m$^{-3}$ (Fig. 6f, yellow line and g). This undersaturation of anthropogenic carbon (Fig. 6f, red bars and h) gives the waters a greater capacity to hold additional $CO_2$ from the atmosphere when they outcrop into the mixed layer (and where thermal and biological processes drive pCO$_2$ levels down in the surface ocean). The effect of this undersaturation in anthropogenic carbon can be represented by the difference between the actual pCO$_2$ (Fig. 6e, red bars) and a hypothetical pCO$_2$ the waters would have if saturated (Fig. 6e, black bars). The undersaturation of anthropogenic carbon leads to a more negative $\Delta$pCO$_2$ that varies from typically −125 µatm in the subtropics to −25 µatm in the subpolar latitudes (Fig. 6i). Hence, the undersaturation of anthropogenic carbon in the waters outcropping in the subpolar gyre enhances the subpolar ocean uptake of atmospheric $CO_2$.

This viewpoint of the Gulf Stream carrying undersaturated anthropogenic carbon following the path of the nutrient stream to the subpolar gyre is consistent with the analysis first provided by Ridge and McKinley[39,40]. Ridge and McKinley additionally argue that this advective supply of waters depleted in anthropogenic carbon is sufficient to account for the area-averaged subpolar ocean uptake of anthropogenic carbon.

### Sensitivity of subpolar carbon to upstream waters

To test our conjecture that the Gulf Stream provides a biogeochemical stream that affects the carbon content of the subpolar North Atlantic, we conduct a formal sensitivity study of an ocean model (referred to as an adjoint model) including a carbon cycle (Methods). An objective function is defined that measures the carbon content over the upper 500 m of the subpolar gyre (marked by red box in Fig. 7). The adjoint model provides the sensitivity of that carbon content to upstream carbon carried in a potential density layer centred on $\sigma_\theta = 27.5$ kg m$^{-3}$. For a lead time of 1 year, the subpolar carbon content is only sensitive to carbon held within its subpolar domain (Fig. 7a). For a lead time of 4 years, there is a band of high sensitivity running along the path of the Gulf Stream and the western side of the subtropical gyre and, for a lead time of 8 years, that high sensitivity band extends from the Gulf Stream to the southwestern side of the subtropics (Fig. 7b, c).

Hence, the model sensitivity maps reveal upstream influence from the Florida Straits and the Gulf Stream extension into the subpolar North Atlantic on lead times of 4 to 8 years. This analysis suggests that the carbon content of the subpolar gyre is influenced by waters originating from the subtropical gyre. These sensitivity maps are also consistent with a similar analysis of how Labrador Sea heat content is sensitive to the upstream sub-surface temperature carried by the Gulf Stream on a decadal timescale[41].

### Discussion and future outlook

Climate projections suggest that the effectiveness of natural ocean carbon sinks in curbing the rise of atmospheric $CO_2$ is likely to diminish in the future[1], based upon on assessment of global ocean carbon-cycle feedbacks[2]. The global ocean response to rising atmospheric $CO_2$ involves competing effects from (i) the rise in atmospheric $CO_2$ altering ocean chemistry and an ocean acidity feedback, and (ii) a smaller and opposing climate change effect, including the temperature-solubility feedback and effects of stratification and circulation changes. For the ocean carbon-cycle feedbacks evaluated over the North Atlantic, maps for the increase in ocean carbon storage[42] reveal local maxima over the Gulf Stream and much of the subpolar gyre (Fig. 8a), which are primarily due to the direct effect of the rising atmospheric $CO_2$ (Fig. 8b). The effect of the climate change leads to reduced ocean carbon uptake over most of the subpolar North Atlantic and Norwegian Sea (Fig. 8c).

The climate feedback on carbon uptake is usually viewed in terms of surface warming decreasing solubility, weakening ocean ventilation and vertical nutrient supply. However, alternatively, this climate feedback may be due to a dynamical response that affects the carbon uptake. In the Atlantic Ocean, this response is connected to changes in the Atlantic Meridional Overturning Circulation[42]. A weakening in the strength of the meridional

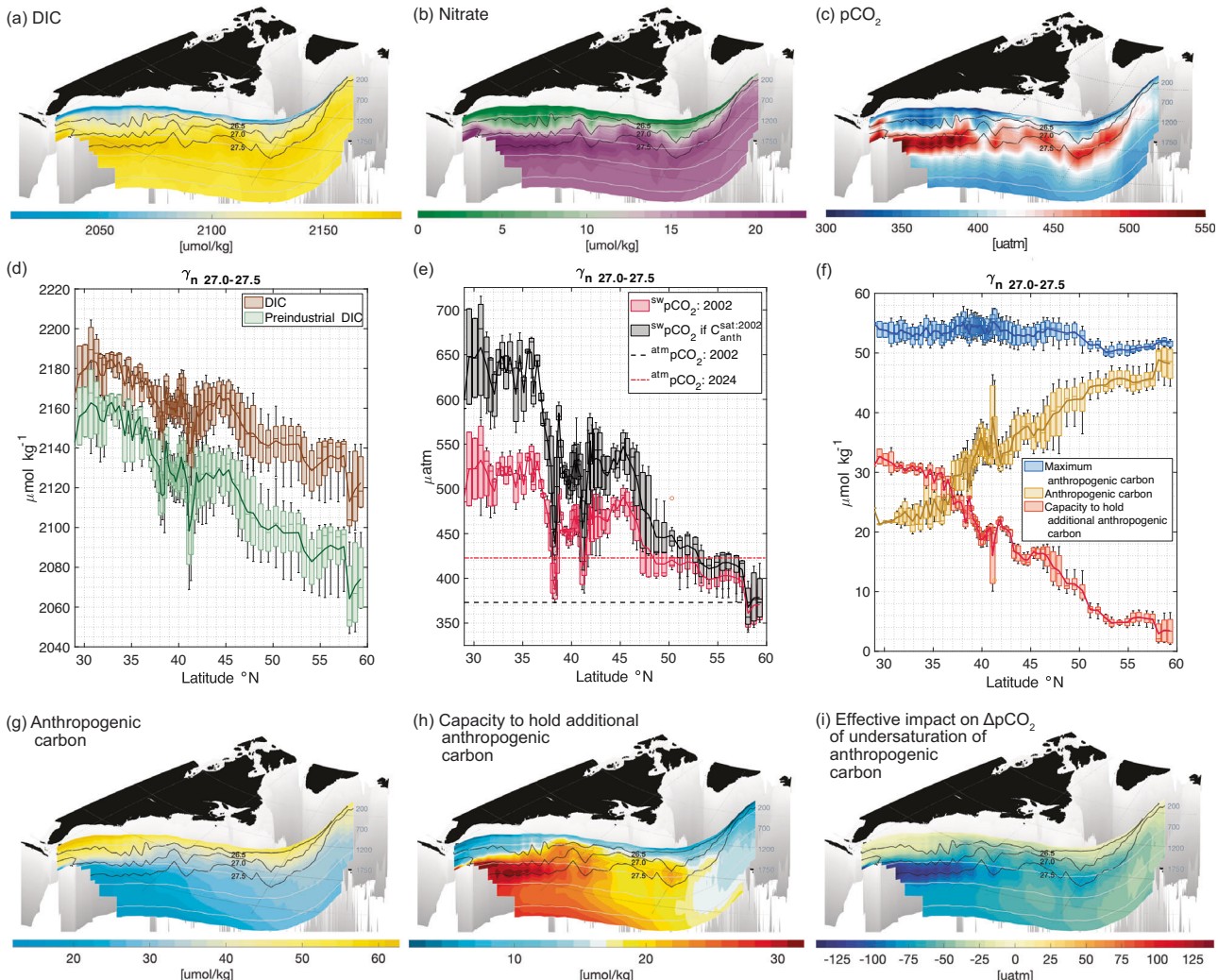

**Fig. 6 | Downstream biogeochemical evolution over the North Atlantic.** Biogeochemical properties along a possible Gulf Stream pathway extending from the subtropical to the subpolar gyre for (**a**) dissolved inorganic carbon (DIC) ($\mu$mol kg$^{-1}$); (**b**) nitrate ($\mu$mol kg$^{-1}$); (**c**) partial pressure of dissolved carbon dioxide (pCO$_2$, $\mu$atm) ; scatterplots of biogeochemical properties along neutral density layers, $\gamma$ for 27.0 to 27.5 kg m$^{-3}$, versus latitude for (**d**) DIC (brown) and pre-industrial DIC (green) ($\mu$mol kg$^{-1}$); (**e**) partial pressure of dissolved carbon dioxide (red) (pCO$_2$, $\mu$atm) and pCO$_2$ if the waters are saturated with the atmosphere (black) together with the atmospheric partial pressure of 373.1 $\mu$atm for year 2002 and 422.8 $\mu$atm for 2024 (grey dashed and red dashed horizontal lines respectively); and (**f**) anthropogenic carbon (yellow), saturated anthropogenic carbon (blue) and capacity to hold additional anthropogenic carbon (red) (all in $\mu$mol kg$^{-1}$) from GLODAPv2 climatology[65]; (**g**) anthropogenic carbon ($\mu$mol kg$^{-1}$); (**h**) the capacity to hold additional anthropogenic carbon ($\mu$mol kg$^{-1}$); and (**i**) the decrease in the partial pressure of dissolved carbon dioxide due to the undersaturation of anthropogenic carbon (evaluated from the difference between the observed pCO$_2$ for 2002 and the calculated pCO$_2$ if the DIC was in equilibrium with an atmosphere for year 2002. This pathway is representative of a deeper trajectory as in Fig. 5c. The sections include neutral density surfaces for 26.5, 27.0 and 27.5 kg m$^{-3}$.

overturning can lead to a weakening in the nutrient stream and a reduction in the delivery of nutrients to the high latitude ocean affecting biological productivity and associated atmospheric CO$_2$ drawdown[8,31], while also leading at depth to an accumulation of nutrients and additional deoxygenation[43]. In a similar manner, a weakening in the meridional overturning circulation is expected to reduce the supply of older waters with depleted anthropogenic carbon (and a capacity to hold much more anthropogenic carbon) to the surface layers. Hence a weakening in the overturning is expected to decrease the uptake of atmospheric CO$_2$ over the subpolar North Atlantic. This interpretation is consistent with the high sensitivity of the subpolar carbon content to the biogeochemical stream provided by the Gulf Stream (Fig. 7b, c). Thus, the regional climate feedback for carbon uptake may be controlled by the strength of the biogeochemical streams over the North Atlantic.

The future response of the Gulf Stream is unclear as part of its transport is associated with the wind-driven gyre circulation and part with the Atlantic Meridional Overturning Circulation. Cable measurements of the Florida

Current transport so far reveal a robust strength over the last 4 decades[44] and a reconstruction of the Atlantic Meridional Overturning Circulation from available observations does not yet reveal any decline over the last 30 years[45]. However, climate model projections suggest that there is likely to be a consistent weakening of the Atlantic Meridional Overturning Circulation over this century[46], although that overturning is projected to continue even with climate extremes[47]. Therefore, climate change is expected to lead to an eventual weakening of the biogeochemical stream[31] associated with the expected weakening of the Atlantic Meridional Overturning Circulation.

Our analysis of the role of the Gulf Stream and its extension has focussed on its basin-scale connections over the North Atlantic. There may though be important smaller scale effects, such as involving sub-mesoscale exchanges[48,49] and mesoscale eddy stirring[50], that affects the downstream dilution of the biogeochemical properties carried by the Gulf Stream and its extension over the North Atlantic (as suggested in Fig. 6a, d). Biogeochemical streams should carry over for other ocean basins; for example, see the review of the biogeochemical effects of the nutrient stream for the

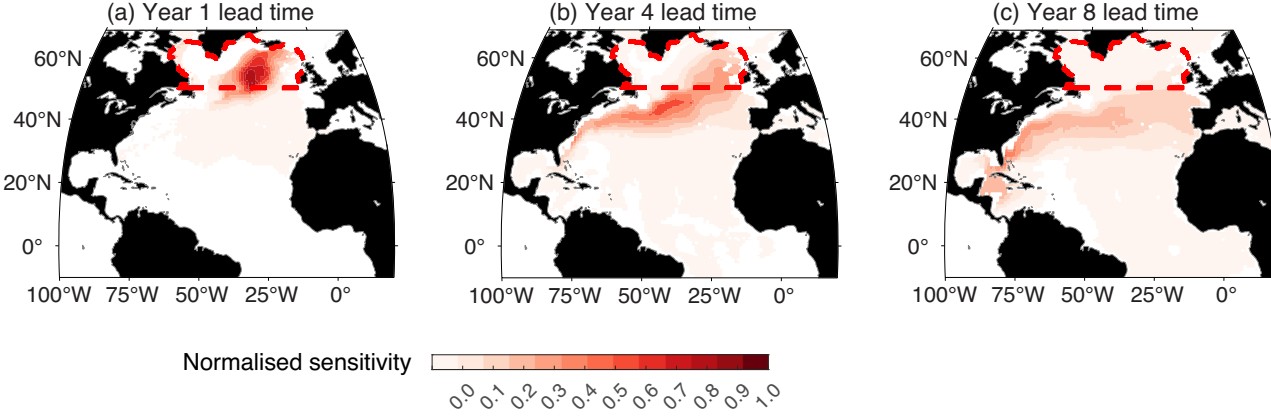

**Fig. 7 | Sensitivity of the ocean carbon storage for the subpolar North Atlantic.** The maps show the normalised sensitivity (Methods) of the volume-averaged, annual mean dissolved inorganic carbon (DIC) concentration in the upper 500 m of the subpolar North Atlantic Ocean (red dashed lines) to upstream DIC anomalies within a potential density layer centred on $\sigma = 27.5$ kg m$^{-3}$ across the entire North Atlantic. Each panel depicts different lead times between the DIC anomaly and the target year of the average: (**a**) approximately lead time of 1 year, (**b**) lead time of 4 years and (**c**) lead time of 8 years. To highlight relative sensitivities, the fields have been scaled by their maximum value.

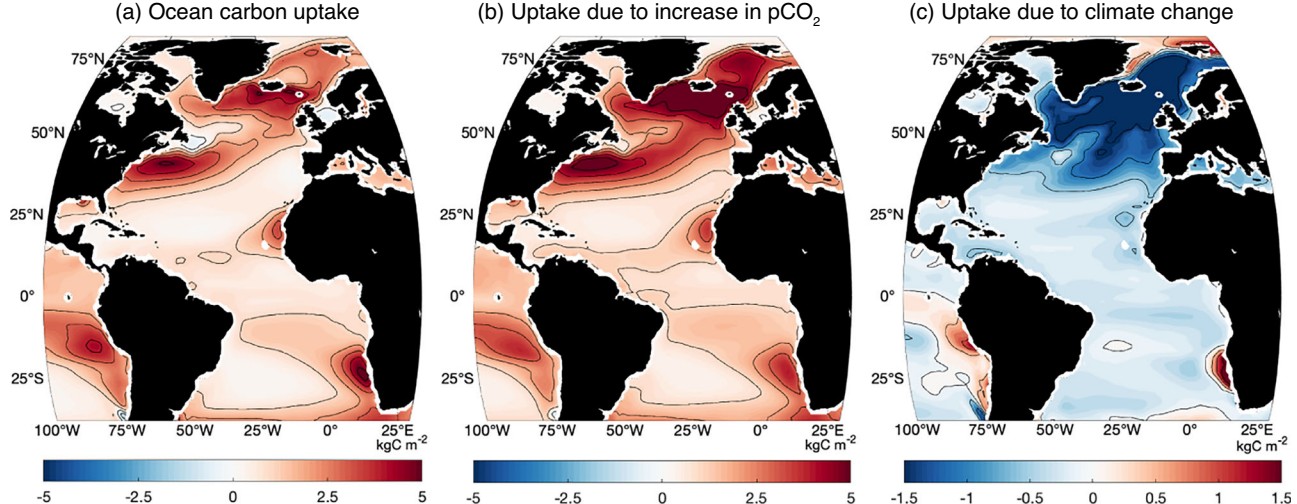

**Fig. 8 | Ocean carbon uptake and feedbacks for idealised climate projection.** Ocean carbon response and feedbacks evaluated using projections of future carbon storage and uptake over the North Atlantic[42]. Maps of the inter-model mean of accumulated ocean carbon uptake (kgC m$^{-2}$) on years 121–140 under a 1%yr$^{-1}$ increase in atmospheric $CO_2$ for 140 years until the $CO_2$ quadruples at year 140: (**a**) change in total ocean carbon uptake; (**b**) change in ocean carbon uptake due to the carbon-concentration response and feedback to an increase in atmospheric $CO_2$; and (**c**) change in ocean carbon uptake due to the carbon-climate feedback. Diagnostics are based on the fully coupled simulation and biogeochemically-coupled simulation for 11 CMIP6 models[2].

Kuroshio in the North Pacific[51]. The biogeochemical streams for the other basins all participate in longer overturning pathways as part of the global overturning circulation[52], so are likely to have higher values of remineralised nutrients and dissolved inorganic carbon.

In summary, the Gulf Stream provides an important contribution to the climate system through its redistribution of heat, nutrients and carbon. The Gulf Stream provides subsurface pathways, redistributing nutrients and older waters undersaturated in anthropogenic carbon over the North Atlantic basin, which may be viewed in terms of a biogeochemical stream. The subpolar carbon content is particularly sensitive to the carbon properties carried in dense layers connecting to the Gulf Stream. When these density layers outcrop into the winter mixed layer, air-sea exchange is affected by the properties carried in the biogeochemical stream, in an analogous manner to how biological production is affected by the supply of nutrients from the nutrient stream[29] (Fig. 3d). Biogeochemical streams are particularly strong in the North Atlantic due to the western boundary current being reinforced by the upper limb of the meridional overturning circulation. The future outlook for the North Atlantic carbon sink is then affected by how climate change alters the strength of the Gulf Stream and the biogeochemical properties carried by the current.

## Methods

The surface signals of the Gulf Stream and its surrounding environment (Fig. 1) are mapped using monthly output for December 2022 from an Operational Mercator global ocean analysis and forecast system[53] at 1/12° horizontal resolution together, with ERA5 air-to-sea net surface heat flux for December 2022 at 1/4° resolution[54] (W m$^{-2}$) and annual nitrate utilisation from the World Ocean Atlas 2023 climatology[55], and sea-to-air $CO_2$ flux at 1° resolution (mol m$^{-2}$yr$^{-1}$) for 2019[56]. The air-sea $CO_2$ flux is calculated according to $F = kK_0(\Delta pCO_2)$, where $F$ is the flux, $k$ is the gas transfer velocity, $K_0$ is the solubility, and $\Delta pCO_2$ is the difference between $pCO_2$ in surface seawater and air. The sea-to-air $CO_2$ flux is further separated into thermal and biophysical components[57,58]: $\Delta pCO_2^{observed} = \Delta pCO_2^{thermal} + \Delta pCO_2^{biophysical}$, where the thermal contribution is derived using an empirical[59] temperature-$pCO_2$ decomposition, $\Delta pCO_2^{thermal} = pCO_{2,ann} \exp(\alpha(T_{month} - T_{ann}))$, and the

effect of biophysical changes is diagnosed as a residual; here *ann* represents annual mean and *month* represents monthly mean, and $\alpha = 0.0423$ K$^{-1}$.

The nutrient and carbon observations across Florida Straits at 27°N are the climatological mean for all hydrographic cruises available in GLODAPv2[60], roughly equivalent to the year 2002. The maximum anthropogenic carbon concentration is calculated as the difference between a dissolved inorganic carbon (DIC) concentration calculated using in situ alkalinity, salinity, temperature, silicate and phosphate data and a pCO$_2$ of 280 $\mu$atm (for the time of the preindustrial) or 373.1$\mu$atm (for the time of the data collected in 2002) using CO2SYS[61]. The potential of waters to absorb additional anthropogenic carbon from the atmosphere, referred to as carbon uptake potential ($\mu$mol kg$^{-1}$), is then estimated as the difference between the maximum anthropogenic carbon concentration and the estimated in situ anthropogenic carbon concentration in 2002. Horizontal property fluxes (in mol m$^{-2}$s$^{-1}$) were calculated by combining climatological property fields calculated from historical hydrographic occupations[60] with a climatological velocity field calculated from historical NOAA cable calibration surveys[44]. Anthropogenic carbon is calculated following the methodology described in Brown et al.[6].

The Gulf Stream pathways are evaluated using virtual seawater particles tracked over a 10-year period (Fig. 5) using the monthly climatological mean three-dimensional flow fields from an ocean state estimate from ECCOv4 release 2[36,62]. 10,000 virtual seawater particles are initially released at Florida Strait in three different depth ranges: 0–110 m, 110–380 m and 380–860 m. The particles are advected by the three-dimensional circulation; the trajectories are evaluated using Drifters.jl and MITgcm.jl[63,64]. To ensure a continual source of particles, every month 2% of the particles are randomly selected and their position reset to Florida Strait.

The estimates of the biogeochemical properties following a subtropical-subpolar pathway (Fig. 6a, b, g) are evaluated using DIC, nitrate and anthropogenic carbon reconstructions from gridded GLODAPv2 climatology[65]. The estimates of ocean pCO$_2$ (Fig. 6c and e, red bars) are along a neutral density layer, 27.0 to 27.5 kg m$^{-3}$, consistent with the DIC distribution and are compared with atmospheric pCO$_2$ of 373.1 $\mu$atm for year 2002 and 422.8 $\mu$atm for year 2024. The ocean pCO$_2$ values are also provided for a hypothetical saturated state in equilibrium with the atmosphere for year 2002 (Fig. 6e, black bars and h).

An adjoint model is used to provide an efficient means of calculating the sensitivity of a chosen ocean model output (the 'objective function') to all input parameters or initial conditions. This approach applies an adjoint operation to the ocean numerical model, effectively converting a 'forward' set of numerical integration steps into a set of 'backward' steps, enabling explicit calculations of how model states depend on earlier steps in the numerical integration. This procedure enables the computation of sensitivities for numerous inputs in a single backwards integration, tracing the influence of upstream properties, surface forcings, and even mixing coefficients on a selected objective function. This calculation is particularly useful for comprehensive sensitivity analyses, as it can be mathematically equivalent to hundreds or even thousands of individual perturbation experiments.

The sensitivity calculations for the subpolar carbon content (Fig. 7) are performed using the adjoint of the ocean model ECCOv4 release 2 (ECCOv4r2)[36,62]. The ECCOv4r2 model is coupled to an idealised ocean carbon cycle model that simulates the cycles of DIC, alkalinity, phosphate, dissolved organic phosphorus, and dissolved oxygen[66–68]. A 500-year spin-up of the model is performed (denoted as ECCOv4r2-DIC) under pre-industrial atmospheric CO$_2$ conditions (284.32 ppm, corresponding to the 1 January 1850 value following biogeochemical protocols[69]). During the 500-year spin-up phase, the detrended atmospheric forcing from the ECCOv4r2 state estimate is applied for a repeated 20 year cycle that is taken for the period 1992-2011; this approach is similar to that used in a previous study[70], apart from detrending the forcing. After the spin-up from year 1850, the atmospheric CO$_2$ concentration was increased following observational records up to year 2011[71]. The resulting ECCOv4r2-DIC simulations from years 1992 to 2011 are used for the carbon-cycle adjoint analyses.

The sensitivity of the carbon content is defined as the change in the annual DIC content, averaged over a selected "control volume" given by the upper 500 m of the subpolar North Atlantic (red dashed line in Fig. 7), in response to hypothetical 14-day scale anomalies in upstream DIC concentrations. The sensitivity fields are scaled and then normalized for each time slice. Firstly, the raw sensitivity, $\partial J/\partial DIC$, is scaled by multiplying by the spatial standard deviation, $\sigma_{DIC}$, of the DIC and then divided by the cost function, $J$, such that $S_J = \frac{\partial J}{\partial DIC} \frac{\sigma_{DIC}}{J}$. Secondly, the scaled sensitivity, $S_J$, is normalized by dividing by its maximum value on that isopycnal from lag year 1, allowing the sensitivities to be compared across different times. The sensitivity contains some small negative values that have a magnitude of only 1% of the maximum positive value so that we focus on the positive values for $S_J$. The sensitivities are mapped in space and time for a density layer centred on $\sigma = 27.5$ kg m$^{-3}$ to reveal the upstream control of the subpolar DIC content.

## Data availability

Maps of properties and air-sea fluxes in Fig. 1 are from available data sets: Sea surface temperature, salinity and height fields are from the Copernicus Global Ocean Physics Analysis and Forecast 1/12° product (10.48670/moi-00016); ERA5 surface heat flux data is from 10.24381/cds.f17050d7; Surface nitrate fields are from World Ocean Atlas 2023; Surface carbon flux fields are from the gridded MPI-SOM-FFN product v2020 (https://www.ncei.noaa.gov/access/ocean-carbon-acidification-data-system/oceans/SPCO2_1982_present_ETH_SOM_FFN.html). Sections of the nutrient stream at 36N in Fig. 2 are available at https://www.bodc.ac.uk/resources/inventories/cruise_inventory/report/6841/. The modelled nutrient stream in Fig. 3 is reported from https://doi.org/10.1029/2010GB003853. Florida Straits biogeochemical fields in Fig. 4 are from the Global Ocean Data Analysis Project's internally-consistent biogeochemical data product, 2023 version https://glodap.info/index.php/merged-and-adjusted-data-product-v2-2023/. Movies of particle trajectories linked to Fig. 5 are available at https://doi.org/10.7910/DVN/XFJPLX. Along-stream' biogeochemical fields in Fig. 6 are from the Global Ocean Data Analysis Project mapped climatological data product), available at https://glodap.info/index.php/mapped-data-product/. The sensitivity fields for ocean carbon storage for the subpolar North Atlantic derived from the ocean circulation model ECCOv4 in Fig. 7 are available at https://doi.org/10.5281/zenodo.17812937. The ocean carbon feedbacks in Fig. 8 evaluated from the CMIP6 simulations in the CMIP6 archive (https://esgf-node.llnl.gov/search/cmip6, World Climate Research Programme, 2021) and further details of the diagnostics are reported in https://doi.org/10.5194/bg-18-3189-2021.

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

## Acknowledgements

This work was supported by a joint UK Natural Environment Research Council and US National Sciences Foundation Large grant NE/W009501/1 (C-Streams), UKRI-NERC NE/Y005287/1 (ROCCA) and UKRI-NERC NE/Y005589/1 (Atlantis). In addiiton, G.F. acknowledges support from NASA awards 80NSSC23K0355 and 1686358, and the Simons Foundation via the CBIOMES programme, and E.M. acknowledge support from the European Union under grant agreement no. 101083922 (OceanICU) and UK Research and Innovation (UKRI) under the UK government's Horizon Europe funding guarantee [grant number 10054454, 10063673, 10064020, 10059241, 10079684, 10059012, 10048179]. Views and opinions expressed are however those of the authors only and do not necessarily reflect those of the European Union or European Research Executive Agency. Neither the European Union nor the granting authority can be held responsible for them.

## Author contributions

R.W. and P.B. conceived the article, and R.W. wrote the article; P.B. completed the carbon analyses and provided figures 1, 4 and 6; E.M. collected the data and produced figure 2; V.R. integrated the isopycnic model and provided figure 3; G.F. conducted the float trajectories for figure 5; Y.T. and D.J. conducted the adjoint experiments for figure 7; A.K. provided the diagnostics of 11 CMIP6 models for figure 8. All authors provided comments on the manuscript.

## Competing interests

The authors declare no competing interests.
