## [Transparent Peer Review file · Communications Earth & Environment]

The biogeochemical stream of the Gulf Stream

Corresponding Author: Professor Richard Williams

Version 0:

Decision Letter:

Dear Professor Williams,

Your manuscript titled "Biogeochemical streams supply nutrients and depleted anthropogenic carbon over the North Atlantic" has now been seen by 4 reviewers, and we include their comments at the end of this message. They find your work of interest, but some important points are raised. We are interested in the possibility of publishing your study in Communications Earth & Environment, but would like to consider your responses to these concerns and assess a revised manuscript before we make a final decision on publication.

We therefore invite you to revise and resubmit your manuscript, along with a point-by-point response that takes into account the points raised. Please highlight all changes in the manuscript text file.

Please submit your point-by-point responses as a separate file, distinct from your cover letter where you can add responses to the Editors' comments that you do not want to be made available to the reviewers. Word files are preferred. We recommend that any figures, tables or graphs that are included in the response to reviewers are also included in the main article or Supplementary Information.

Please use the following link to submit your revised manuscript, point-by-point response to the referees' comments (which should be in a separate document to any cover letter), a tracked-changes version of the manuscript (as a PDF file) and the completed checklist:

Link Redacted

We hope to receive your revised paper within six weeks; please let us know if you aren't able to submit it within this time so that we can discuss how best to proceed. If we don't hear from you, and the revision process takes significantly longer, we may close your file. In this event, we will still be happy to reconsider your paper at a later date, as long as nothing similar has been accepted for publication at Communications Earth & Environment or published elsewhere in the meantime.

Please do not hesitate to contact us if you have any questions or would like to discuss these revisions further. We look forward to seeing the revised manuscript and thank you for the opportunity to review your work.

Best regards,

Alice Drinkwater, PhD
Associate Editor
Communications Earth & Environment
Consulting Editor
Communications Sustainability

EDITORIAL POLICIES AND FORMATTING

Editorial Policy: [Policy requirements](https://www.nature.com/documents/nr-editorial-policy-checklist.pdf) (Download the link to your computer as a PDF.)

- Behavioural and social science
- Ecological, evolutionary & environmental sciences
- Life sciences

<https://www.nature.com/documents/nr-reporting-summary.zip>

Furthermore, please align your manuscript with our format requirements, which are summarized on the following checklist: [Communications Earth & Environment formatting checklist](https://www.nature.com/documents/commsj-phys-style-formatting-checklist-article.pdf)

and also in our style and formatting guide [Communications Earth & Environment formatting guide](https://www.nature.com/documents/commsj-phys-style-formatting-guide-accept.pdf) .

*** DATA: Communications Earth & Environment endorses the principles of the Enabling FAIR data project (<http://www.copdess.org/enabling-fair-data-project/>). We ask authors to make the data that support their conclusions available in permanent, publically accessible data repositories. (Please contact the editor if you are unable to make your data available).

All Communications Earth & Environment manuscripts must include a section titled "Data Availability" at the end of the Methods section or main text (if no Methods). More information on this policy, is available at <http://www.nature.com/authors/policies/data/data-availability-statements-data-citations.pdf>.

If a community resource is unavailable, data can be submitted to generalist repositories such as [figshare](https://figshare.com/) or [Dryad Digital Repository](http://datadryad.org/). Please provide a unique identifier for the data (for example a DOI or a permanent URL) in the data availability statement, if possible. If the repository does not provide identifiers, we encourage authors to supply the search terms that will return the data. For data that have been obtained from publically available sources, please provide a URL and the specific data product name in the data availability statement. Data with a DOI should be further cited in the methods reference section.

REVIEWER COMMENTS:

Reviewer #1 (Remarks to the Author):

The manuscript made two fatal mistakes:

1. There is a mix-up of CO₂ and anthropogenic CO₂. The manuscript claims, "These older waters with low anthropogenic carbon can be viewed as having a high potential to uptake anthropogenic carbon when they reach the surface". The fact is that whether the surface water picks up CO₂ depends on the pCO₂ difference between the surface water and the atmosphere and has nothing to do with whether the surface water contains anthropogenic CO₂. Statements related to the

wrong concept can be found in the Abstract and throughout the manuscript.

2. The second fatal error was to assume that when the high nutrients in the Nutrient Stream reach the surface ocean, the enhanced biological productivity would enhance the uptake of anthropogenic CO₂. As stated above, even when the enhanced biological productivity reduces the surface pCO₂ to below saturation, the surface ocean would pick up CO₂ regardless of whether the water contains anthropogenic CO₂. However, it has already been shown that when the Kuroshio nutrient stream upwells to the euphoric layer, it does not necessarily pick up CO₂. This is because waters at the depth of the nutrient stream contain over-saturated CO₂, so they tend to release CO₂ when reaching the surface. At the end of the day, whether there is uptake due to enhanced biological productivity or release of CO₂ due to oversaturation depends on the N/DIC and P/DIC ratios (Pan et al., *Science of the Total Environment*, 511, 692-702, 2015; Chen, et al., *Scientific Reports*, 11, article 5080, 2021)

Reviewer #2 (Remarks to the Author):

Please find enclosed a file with my comments to the authors.

I'm happy to sign this review, Josep L. Pelegrí.

Reviewer #3 (Remarks to the Author):

The manuscript is well written, of broad interest, and while the main thrust of the article, that the Gulf Stream is a "biogeochemical stream" already exists in various forms in the literature (as shown in the introductory review here), in this reviewer's opinion the role of the GS as a biogeochemical stream is not well understood or properly emphasized in the literature and is well deserving of being reviewed and highlighted.

Some specific comments:

The density layer where you show the highest nitrate flux (26.5-27.5) does appear to outcrop intermittently well south of the subpolar gyre. This has been shown in some previous studies such as <https://doi.org/10.1029/2023JC020526> and <https://doi.org/10.1002/2015GL067152>. It may be worth including something about this to note if it is minor compared to your overall points, or another factor in the full biogeochemical portfolio of the Gulf Stream, and a factor that very likely would be missing in coarser resolution modeling.

Related, in this line "This outcropping then transfers the nutrients into the winter mixed layer, ultimately helping to sustain high-latitude biological productivity." It is not clear to that this nutrient strata is only outcropped into the winter mixed layer, is it too deep in the summer to have sufficient light for photosynthesis? It seems likely that it may have sufficient light for growth in some regions even at depth. Can you provide a sentence or two of background as to why it does/doesn't shoal more consistently or whether there is enough light for this layer to grow during the summer? I mention this because there does seem to be some evidence that it might shoal intermittently all year, but the total BGC impact of this ephemeral shoaling may be negligible compared to the injection of nutrients into the winter mixed layer.

Figure 5 and the particle tracking experiments are compelling for the argument presented here and are nicely presented. Are these supported by any observational work and can you add some discussion on this?

The text on the figures needs to be larger, it is challenging to read, particularly Fig 4 I cannot read the text even when zooming in due to low resolution and small font.

The line "Hence, the carbon content of the subpolar North Atlantic is sensitive to the advection of carbon anomalies from the Gulf Stream and its extension" is possibly too certain given the one experiment described and lack of observational evidence. While the modeling is very interesting and I am not too skeptical of your result, it may be more appropriate to say "This modeling suggests the carbon content of the subpolar North Atlantic is sensitive to the advection of carbon anomalies from the Gulf Stream and its extension..."

The first two paragraphs of the "Discussion" section should be re-written for clarity. First you note that natural ocean carbon sinks are projected to diminish, but then say that the ocean carbon uptake is projected to increase. By "natural" do you mean biological? I don't think so, or do you mean that their efficiency will decrease? but not the total amount of carbon taken up because of increasing atmospheric pCO₂? I may be missing a nuance, but please clarify this a bit.

In the second to last line of the manuscript "Biogeochemical streams should carry over for other ocean basins..." it might be helpful to readers to be a bit more explicit when discussing the generalizability of your ideas on the Gulf Stream as a BGC stream. This could be a useful reference in pointing to the Kuroshio: <https://doi.org/10.1002/9781119428428.ch6>.

Overall a well written, accessible manuscript that I expect will be of interest to a broad audience.

Reviewer #4 (Remarks to the Author):

Previous studies that focus on the effects of increased atmospheric CO₂ and warming of ocean surface, reducing carbon uptake potential by the ocean, are ignoring the importance of western boundary currents, such as Gulf Stream, that carry the potential of anthropogenic carbon uptake in its subsurface layers toward downstream regions where these waters are outcropped.

A lack of pathway from subtropical to subpolar gyre near the surface has been remarked in other publications due to Ekman processes. However, this study presents the trajectories of Lagrangian particles released at different depth levels, showing the importance of a biogeochemical stream that extends from subtropical gyre to subpolar gyre in denser layers. The further analysis showed that subpolar DIC is largely controlled by the upstream DIC through the Gulf Stream. This subsurface transport of high carbon uptake potential by the Gulf Stream, which has not been highlighted, is very important for better predicting climate change. Also, the effect of the biogeochemical stream is not limited to local region, but it could provide large impacts at global scale since the subpolar North Atlantic is the major CO₂ sink for the atmosphere. Therefore, in my opinion, the paper can be a nice contribution to the journal, *Communications, Earth and Environment*, after some revisions for several points as follows.

Major Comments

- Please add the line numbers.
- My focus is not biogeochemical process, yet it is very nice to know that the Gulf Stream is controlling at some level the subpolar North Atlantic DIC. Does this also mean that the high DIC transported in the subsurface Gulf Stream is supplying negative total carbon uptake potential downstream as opposed to the anthropogenic carbon uptake potential? Please clarify how the carbon uptake potential provided by the Gulf Stream works in net carbon uptake, not only limited to the anthropogenic part. This is very important information for readers outside the scope of biogeochemistry.
- While solubility pump is mentioned, biological pump is not discussed in the manuscript at all. The manuscript showed the importance of lateral transport and shoaling of high carbon uptake potential toward the downstream of the Gulf Stream. Yet, it is not clear to me how these two pumps work in the atmospheric CO₂ uptake.
- In the Introduction it is indicated that Gulf Stream contribution to subpolar biological production is not well-recognized 'The Gulf Stream is a major part of the upper limb of the Atlantic Meridional Overturning Circulation, [...] its contribution to mitigating human-driven atmospheric carbon increases and sustaining subpolar biological productivity is not'. However, the part of sustainability of biological production to higher latitudes is relatively well-known by in bibliographies in the manuscript such as references [31, 33]. Perhaps, it would be better to review the previous studies (some of the studies are actually by the authors) more precisely.
- The following part in the Introduction sounds it doesn't belong here since this paragraph is about previous studies that are not directly related to the present study.
"There is an ongoing debate about the importance of this ocean heat release to the atmosphere in determining the relatively mild winter climate of western Europe^{23–26}. Contrary to widespread perception, there is a view that the Gulf Stream is unimportant in explaining how western Europe is warmer in winter than an equivalent latitude on the eastern side of North America²³. Instead, the warming of western Europe is explained by the atmospheric wave pattern set up by orography and seasonal heat release from the ocean to the atmosphere during winter. However, this view is challenged by an argument that seasonal heat release from the local ocean is insufficient to explain the warming and needs to be augmented by the heat supply from ocean transport involving the Gulf Stream²⁴. Alternatively, there is an intermediate view that warming from the ocean generates an atmospheric wave pattern leading to cold air on the eastern seaboard of North America and warmer air over western Europe and, thus, explains the winter temperature contrast between western and eastern continental boundaries²⁷".

Minor Comments

- It is pointed out during winter season (reference 28) the potential of carbon uptake by the ocean (Gulf Stream), what about other seasons? Is there (un)published information about similar potential in other time of the year? Although this is mainly for heat fluxes.
- Figure 4 is a little confusing, as (a) belongs to 2019-2021, (b) from 1998 occupation, (c) corresponds similar to b I guess? And as for (e-f), values of b&c are combined with velocity values from 2000-2020 from (d)? Although in the manuscript it is clear the relation of the current and the potential outcropping of subsurface waters, including nitrate and carbon uptake potential, the period where the values are averaged are a little confusing. Also, the numbers and letters in Figure 4 are quite small (need to zoom in a lot to carefully read the values in isopycnals and axis).
- Sadly, the link to the video of Additional information is not available anymore.
- In Method: "... 1998 property field with average water velocities"
This needs "." at the end. Also, please make the explanation better to be easy to understand for non-specialist, why taking this difference in DIC means anthropogenic carbon uptake.
- In Method: "Every month 2% of the particles are randomly selected and their position reset to Florida Strait".
Why randomly selected 2% of the particles only are relocated to the Florida Strait? Please elaborate on this. Also, is vertical mixing considered in the particle tracking? Please justify it if it is not.

Communications Earth & Environment is committed to improving transparency in authorship. As part of our efforts in this direction, we are now requesting that all authors identified as 'corresponding author' create and link their Open Researcher and Contributor Identifier (ORCID) with their account on the Manuscript Tracking System prior to acceptance. ORCID helps the scientific community achieve unambiguous attribution of all scholarly contributions. You can create and link your ORCID from the home page of the Manuscript Tracking System by clicking on 'Modify my Springer Nature account' and following the instructions in the link below. Please also inform all co-authors that they can add their ORCIDs to their accounts and that they must do so prior to acceptance.

Version 1:

Decision Letter:

Dear Professor Williams,

Your manuscript titled "The biogeochemical stream of the Gulf Stream" has now been seen by our reviewers, whose comments appear below. In light of their advice we are delighted to say that we are happy, in principle, to publish a suitably revised version in Communications Earth & Environment.

We therefore invite you to revise your paper one last time to address the remaining concerns of our reviewers, which pertains to clarifying the uptake of non-anthropogenic carbon dioxide. At the same time we ask that you edit your manuscript to comply with our format requirements and to maximise the accessibility and therefore the impact of your work.

EDITORIAL REQUESTS:

*****Please take care to match our formatting and policy requirements. We will check revised manuscript and return manuscripts that do not comply. Such requests will lead to delays. *****

SUBMISSION INFORMATION:

OPEN ACCESS:

Communications Earth & Environment is a fully open access journal. Articles are made freely accessible on publication. For further information about article processing charges, open access funding, and advice and support from Nature Portfolio, please visit <https://www.nature.com/commsenv/open-access>

Link Redacted

Best regards,

Alice Drinkwater, PhD
Associate Editor
Communications Earth & Environment
Consulting Editor
Communications Sustainability

REVIEWERS' COMMENTS:

Reviewer #1 (Remarks to the Author):

The authors still do not understand the difference between CO₂ and the anthropogenic CO₂, which is only about 3% of the total CO₂. The two primary "carbon pumps", namely the biological pump and the physical pump, move carbon regardless of whether it is natural or anthropogenic. Much of the description of the anthropogenic carbon is wrong. For instance, they compare the waters' anthropogenic carbon concentration with the modern atmosphere. This comparison has nothing to do with whether the water takes up carbon, anthropogenic or not.

A second issue is whether upwelling leads to uptaking or degassing of CO₂. I used the Kuroshio example to show that upwelling may or may not lead to degassing. The authors are correct that their study area is the Gulf Stream, but no similar study appears to exist in the Gulf Stream. Note that uptaking or degassing of CO₂ has nothing to do with whether the CO₂ is anthropogenic.

Reviewer #2 (Remarks to the Author):

Second Review of "The biogeochemical stream of the Gulf Stream", formerly entitled "Biogeochemical streams supply nutrients and depleted anthropogenic carbon over the North Atlantic", by R. G. Williams et al., submitted to Communications Earth & Environment

This is a second review of the manuscript entitled "The biogeochemical stream of the Gulf Stream", formerly entitled "Biogeochemical streams supply nutrients and depleted anthropogenic carbon over the North Atlantic", by R. G. Williams et al. As exposed in my original review, I believe this is a timely manuscript that looks at the relevance of biogeochemical processes in the Gulf Stream. In this new version, the authors have done a great job, I would like to thank them for their efforts and congratulate them for the outcome. In particular, they have addressed a main issue raised in my earlier review, namely they have provided a much better description of the along-stream changes in both total dissolved inorganic carbon (DIC) and its anthropogenic fraction, which is necessary to better elucidate what will be the ocean carbon uptake caused by an increase in atmospheric CO₂.

In their response to my earlier review, the authors have also included several comments on which I may disagree, but these do not affect substantially this new manuscript version so I will only refer to one specific issue. Palter and Lozier (2008) used an oxygen-corrected phosphate tracer to show that intermediate layers (potential density in the range 26.0 to 27.0) have a >80% subantarctic contribution, but this is not the same as saying that 80% of the nutrient load in these layers is of subantarctic origin (line 76 of the manuscript). Figure 3c, in my opinion, does not evidence this western-boundary transport continuity from the tropics to the subtropical North Atlantic.

I have only a couple of additional minor suggestions for the authors' consideration:

- 1) In my opinion, repeating the word "stream" in a seven-words title sound rare. I prefer not to make any suggestion but I leave it to the authors, and the editor, for consideration.
- 2) In line 90 it is said that about half of the nutrient concentrations are preformed and half remineralized, referring to figure 4. However, the results for nitrate flux in Figures 4e,f suggest that nitrate preformed flux is relatively small as compared to the total.

I would like to congratulate the authors for a nice piece of research, I'm happy to recommend publication in the journal.

Josep L. Pelegrí

Reviewer #3 (Remarks to the Author):

Overall I am quite satisfied with the authors' responses to the issues raised in the last review. The additional review and discussion helps clarify the manuscript's main points.

While it may be beyond the main scope of this work, I would note in regard to the low light level point, that many field programs are increasingly finding photosynthesis occurring at extremely low light levels and this may play a role here as in: <https://www.nature.com/articles/s41467-024-51636-8>. But it is a good point that if these waters stay below the summer mixed layer the effect will be delayed into the winter.

The notes and increased description of the sensitivity analysis are helpful. And while the adjoint model of course effectively represents many perturbations in the system, there is an underlying assumption that the model architecture (and the parameters and parameterizations of that model) can effectively represent this system, which may or may not be true and this is why I suggested the more cautious phrasing. But regardless, I accept the authors' preference for their current phrasing and believe it is reasonable.

Some figures are still fairly low resolution (e.g. the x and y axis labels and contour labels on Figure 4 are very low resolution even though the subplot titles are high res) but I imagine this can be quickly fixed in the final proofing process.

I look forward both to seeing this work published and to reading your future reports on the field program you describe.

Reviewer #4 (Remarks to the Author):

The authors have revised all questions

** Visit Nature Portfolio's author and referees' website at <http://www.nature.com/authors> for information about policies, services and author benefits**

REVIEWER COMMENTS:

Reviewer #1 (Remarks to the Author):

The manuscript made two fatal mistakes:

Thank you for the comments. We have modified the text to clarify our viewpoint and address the points raised by the referee.

1. There is a mix-up of CO₂ and anthropogenic CO₂. The manuscript claims, "These older waters with low anthropogenic carbon can be viewed as having a high potential to uptake anthropogenic carbon when they reach the surface". The fact is that whether the surface water picks up CO₂ depends on the pCO₂ difference between the surface water and the atmosphere and has nothing to do with whether the surface water contains anthropogenic CO₂. Statements related to the wrong concept can be found in the Abstract and throughout the manuscript.

We agree that our viewpoint requires further expansion including details of how the natural carbon varies. The referee is correct that air-sea exchange of carbon depends upon the $\Delta p\text{CO}_2$ between the atmosphere and surface ocean. We now include the $\Delta p\text{CO}_2$ difference in our analyses. Our perspective sets out a view as to how an upstream control affects the $\Delta p\text{CO}_2$ rather than being determined solely by local processes.

In particular, we include a distribution of DIC and $\Delta p\text{CO}_2$ along density layers in the subtropical Atlantic (Fig. 6a,d, Fig. S4) and examine the potential for these waters to take up anthropogenic carbon (by comparing the waters' anthropogenic carbon concentration with the modern atmosphere).

Scatterplots of pCO₂ reveals a progressive decrease with latitude for a dense layer, 27 to 27.5 kg m⁻³ (Figure S4c). When that pCO₂ in the layer is compared with the atmospheric pCO₂ in 2002, then there is either a slight outgassing or no air-sea uptake (in accord with the arguments of the referee). However, if you compare the pCO₂ in the layer with the atmospheric pCO₂ in 2024, then there is an implied ocean uptake of CO₂ over the subpolar latitudes (due to the progressive rise in atmospheric CO₂).

We demonstrate that the undersaturation of anthropogenic carbon affects that $\Delta p\text{CO}_2$ signal. In Figure 6c-f, we show the actual $\Delta p\text{CO}_2$ for 2002 (Fig. 6d) and how $\Delta p\text{CO}_2$ would be higher if the water column was saturated in anthropogenic carbon (Fig. 6e), so that by comparing with the actual $\Delta p\text{CO}_2$ for 2002, we find that the undersaturation in anthropogenic carbon leads to a more negative $\Delta p\text{CO}_2$ by -25 to -125 μatm (Fig. 6f). Thus, the undersaturation in anthropogenic carbon affects the sign of the $\Delta p\text{CO}_2$ over the subpolar gyre for the present day, and so affects the air-sea flux of CO₂ over the subpolar gyre.

In summary, in accord with Ridge and McKinley (2020), we show that the Florida Current provides waters with high potential to uptake anthropogenic carbon. That downstream delivery of waters by the Gulf Stream with high potential to uptake anthropogenic carbon then contributes to the $\Delta p\text{CO}_2$ signal over the subpolar gyre, leading to air-sea uptake of CO₂ (as in Fig. 1c).

We provide additional support for our viewpoint:

(1) An expanded figure 4 for the Florida Straits shows plots for the reconstruction of preindustrial dissolved inorganic carbon (DIC), present day DIC, the anthropogenic carbon (from the mismatch of present day DIC and preindustrial DIC), the saturated anthropogenic carbon and the capacity to uptake anthropogenic carbon (from the mismatch between the saturated anthropogenic carbon and the anthropogenic carbon). The transports of the DIC, the anthropogenic carbon and the capacity to take up additional anthropogenic carbon are also included.

- (2) A new figure 6 showing the downstream evolution of DIC, anthropogenic carbon and the capacity to take up additional anthropogenic carbon over the North Atlantic for the crucial neutral density layer 27 to 27.5 kg m⁻³ together with the $\Delta p\text{CO}_2$ for that layer for 2002 and hypothetical estimates of $\Delta p\text{CO}_2$ if the anthropogenic carbon was saturated.
- (3) Scatter plots of the data (in supplementary figure S4) used for those plots reveals that saturated anthropogenic carbon (blue dots) is relatively constant in this neutral density layer (27 to 27.5 kg m⁻³) and there is a capacity to uptake additional anthropogenic carbon at the Florida Straits that decreases downstream in the basin (yellow dots) as anthropogenic carbon is taken up by the water column (red dots).

New text: L118-142

A biogeochemical stream is proposed following a sub-surface pathway from the subtropics to the subpolar latitudes (consistent with Fig. 5c). A reconstruction of that biogeochemical pathway from climatology reveals surface-depth property contrasts in the subtropics that weaken over the subpolar latitudes (Fig. 6a). Sub-surface waters have high concentrations of DIC in the subtropics, particularly along neutral densities 26.5 to 27.5 kg m⁻³, and there is a downstream decrease in their concentration as density layers shoal and outcrop in the subpolar gyre (Figs. 6a and S4a). Sub-surface waters likewise change from having low concentrations of anthropogenic carbon in the subtropics to progressively higher concentrations in the subpolar gyre (Figs. 6b and S4b). Accordingly there is a high potential for uptake of anthropogenic carbon in the subtropics to lower potential for uptake in the subpolar gyre, particularly along the neutral density layer 27 to 27.5 kg m⁻³, as those waters take up more anthropogenic carbon at the sea surface (Figs. 6c and S4b). This transport of DIC affects the difference in the partial pressure of the dissolved CO₂ and the atmospheric partial pressure

This transport of DIC affects the difference in the partial pressure of the dissolved CO₂ and the atmospheric partial pressure $p\text{CO}_2$, represented by $\Delta p\text{CO}_2$, which is evaluated for year 2002 to be consistent with the timing of the ocean data. There are high values of $\Delta p\text{CO}_2$ along neutral densities 26.5 to 27.5 kg m⁻³ in the subtropics (typically 80 to 150 μatmos for year 2002, Methods), suggesting that if those waters were locally brought to the surface, then at that time there would be an outgassing of CO₂ (Figs. 6d and S4c). There is a downstream dilution of the DIC signal along the same density layer that leads to the subpolar $\Delta p\text{CO}_2$ being much smaller (typically ranging from 17 to 44 μatmos when evaluated for year 2002). Our reconstruction of the $\Delta p\text{CO}_2$ signal is sensitive to the timing given the ongoing rise in atmospheric CO₂ (with an atmospheric increase of 50 μatmos between year 2002 and 2024). The subpolar values of $\Delta p\text{CO}_2$ change from being positive for year 2002 to negative for year 2024 (typically ranging from -6 to -33 μatmos for year 2024) (Fig. S4c). Hence, there is an implied subpolar ocean uptake of CO₂ from the atmosphere for the present day.

The undersaturation of anthropogenic carbon affects this $\Delta p\text{CO}_2$ signal and resulting air-sea flux. For example, if there is a saturation of anthropogenic carbon, then there would be a higher DIC value and associated $\Delta p\text{CO}_2$ (Fig. 6e). The effect of the undersaturation of anthropogenic carbon on the $\Delta p\text{CO}_2$ is represented here by the difference in the actual $\Delta p\text{CO}_2$ and this hypothetical value when there is saturation: this effect varies from typically -125 μatmos in the subtropics to -25 μatmos in the subpolar latitudes using an atmospheric $p\text{CO}_2$ for year 2002 (Fig. 6f). Hence, the undersaturation of anthropogenic carbon is significant in determining the sign of the $\Delta p\text{CO}_2$ signal over the subpolar latitudes (Figs. 6d and S4c).

New text in Methods

L220 to 224

The estimates of the biogeochemical properties following a subtropical-subpolar pathway (Fig. 6a-c) are evaluated using DIC and anthropogenic carbon reconstructions from gridded GLODAPv2 climatology60. The estimates of $\Delta p\text{CO}_2$ (Fig. 6d-f) are from a difference between (i) ocean $p\text{CO}_2$ along a neutral density layer, 27.0 to 27.5 kg m⁻³, consistent with the DIC distribution and (ii) atmospheric $p\text{CO}_2$ of 373.1 μatmos for year 2002 and 422.8 μatmos for year 2024; the values

reported in the text are based upon a 8° running latitudinal mean of the ocean $p\text{CO}_2$ values in Fig. S4c.

Updated Figure 4 including sections of DIC and anthropogenic carbon.

Figure 4. Biogeochemical Stream in the Florida Straits at 27°N : (a) salinity (psu) (with white contours for 12°C and 24°C); (b) nitrate concentration ($\mu\text{mol kg}^{-1}$); and (c) preformed nitrate ($\mu\text{mol kg}^{-1}$); (d) meridional velocity (m s^{-1}); (e) meridional nitrate transport (kmol s^{-1}); (f) meridional preformed nitrate transport (kmol s^{-1}); (g) dissolved inorganic carbon (DIC) ($\mu\text{mol kg}^{-1}$); (h) anthropogenic carbon ($\mu\text{mol kg}^{-1}$); (i) capacity of waters to absorb additional anthropogenic carbon (anthropogenic carbon uptake potential, $\mu\text{mol kg}^{-1}$); (j) northward DIC transport (kmol s^{-1}); (k) northward anthropogenic carbon transport (kmol s^{-1}); and (l) northward anthropogenic carbon uptake potential transport (kmol s^{-1}). The transports are calculated by combining property fields with average water velocities from a series of hydrodynamical cruises in 2012^{38,39} and the plots include neutral density contours 26.5 to 27.5 kg m^{-3} .

Updated Figure 6 including plots of DIC, anthropogenic carbon and $p\text{CO}_2$.

Figure 6. Biogeochemical properties along a possible Gulf Stream pathway extending from the subtropical to the subpolar gyre for (a) dissolved inorganic carbon (DIC) ($\mu\text{mol kg}^{-1}$), (b) anthropogenic carbon ($\mu\text{mol kg}^{-1}$) and (c) anthropogenic carbon uptake potential ($\mu\text{mol kg}^{-1}$) from GLODAPv2 climatology⁴⁵. The associated air-sea difference in partial pressure of dissolved carbon dioxide ($\Delta p\text{CO}_2$, μatm) linked to the DIC distribution is shown for year 2002 in (d) together with (e) the $\Delta p\text{CO}_2$ distribution if the anthropogenic carbon is fully saturated with the atmosphere and (f) the difference in the $\Delta p\text{CO}_2$ between (d) and (e), representing the effect of the under saturation in anthropogenic carbon in (c), where the atmospheric $p\text{CO}_2$ for 2002 is $373.1 \mu\text{atm}$. This pathway is representative of a deeper trajectory as in Fig. 5c. The sections include neutral density surfaces for 26.5 , 27.0 and 27.5 kg m^{-3} .

New supplementary figure S4 including data points for the DIC, anthropogenic carbon and $p\text{CO}_2$ values along a neutral density layer.

Figure S4. Scatterplots of biogeochemical properties along neutral density layers, σ_n for 27.0 to 27.5 , versus latitude along a possible Gulf Stream pathway extending from the subtropical gyre into the subpolar gyre (as in Fig. 6) for (a) DIC (blue dots) and pre-industrial DIC (red triangles) ($\mu\text{mol kg}^{-1}$); (b) anthropogenic carbon (red triangles), saturated anthropogenic carbon (blue dots) and capacity to hold additional anthropogenic carbon (yellow crosses) (all in $\mu\text{mol kg}^{-1}$, blue line); and (c) partial pressure of dissolved carbon dioxide ($p\text{CO}_2$, μatm) together with the atmospheric partial pressure for years 2002 and 2024 (black dashed and red dashed horizontal lines respectively); from GLODAPv2 climatology⁴⁵.

2. The second fatal error was to assume that when the high nutrients in the Nutrient Stream reach the surface ocean, the enhanced biological productivity would enhance the uptake of anthropogenic CO₂.

We do not argue that biological productivity controls the enhanced uptake of anthropogenic carbon, which is instead controlled by the delivery of waters with the capacity to uptake anthropogenic carbon.

We separate the nutrients into preformed and regenerated nutrients, see revised Figure 4b,c that includes regenerated nutrients (also supplementary figure S3 for profiles).

The delivery of regenerated nutrients will not lead to a net effect on the air-sea uptake of carbon dioxide: there will be an associated outgassing from delivery of carbon rich water, but also accompanying biological drawdown.

However, the delivery of preformed nutrients to the euphotic zone will be associated with an uptake of carbon dioxide.

We provide support for this view via Figure 1c-d revealing a subpolar uptake of atmospheric CO₂ that includes both thermal and biophysical contributions. There is a signal of biological productivity from the seasonal utilisation of nutrients (Fig. 1c) that is broadly consistent with the patterns of the biophysical contribution for the uptake of atmospheric CO₂ (Fig. 1f) (although each involve independent data diagnostics). In our view, the delivery of nutrients and enhancement of biological productivity is a contributing factor to the uptake of CO₂ for the subpolar North Atlantic.

New text:

L23-26 *The air-sea heat loss is a consequence of the northward delivery of warm, surface waters and the biological production results from the northward supply of nutrients⁸, and both are affected by the Atlantic Meridional Overturning Circulation.*

L57-58 *such as on basin-scale patterns in seasonal nutrient utilisation (Fig. 1c) and annual air-sea CO₂ exchange over the North Atlantic (Fig. 1d), which includes both thermal and biological contributions (Fig. 1e,f).*

As stated above, even when the enhanced biological productivity reduces the surface pCO₂ to below saturation, the surface ocean would pick up CO₂ regardless of whether the water contains anthropogenic CO₂.

Correct, we do not disagree with that statement. However, the undersaturation of anthropogenic CO₂ is a factor in determining the surface pCO₂ (as supported by diagnostics in Figure 6c-f).

However, it has already been shown that when the Kuroshio nutrient stream upwells to the euphotic layer, it does not necessarily pick up CO₂. This is because waters at the depth of the nutrient stream contain over-saturated CO₂, so they tend to release CO₂ when reaching the surface. At the end of the day, whether there is uptake due to enhanced biological productivity or release of CO₂ due to oversaturation depends on the N/DIC and P/DIC ratios (Pan et al., *Science of the Total Environment*, 511, 692-702, 2015; Chen, et al., *Scientific Reports*, 11, article 5080, 2021)

We agree that there is a delicate balance affecting the pCO₂ distribution and air-sea exchange in subpolar latitudes. For our North Atlantic diagnostics, the water properties in the neutral density layer 27 to 27.5 kg m⁻³ also imply weak outgassing when evaluated in 2002, but imply an air-sea uptake in 2024 (Figure S4c).

New text L131-136

There is a downstream dilution of the DIC signal along the same density layer that leads to the subpolar $\Delta p\text{CO}_2$ being much smaller (typically ranging from 17 to 44 μatmos when evaluated for year 2002). Our reconstruction of the $\Delta p\text{CO}_2$ signal is sensitive to the timing given the ongoing rise in atmospheric CO₂ (with an atmospheric increase of 50 μatmos between year 2002 and 2024). The subpolar values of $\Delta p\text{CO}_2$ change from being positive for year 2002 to negative for year

2024 (typically ranging from -6 to $-33 \mu\text{atmos}$ for year 2024) (Fig. S4c). Hence, there is an implied subpolar ocean uptake of CO_2 from the atmosphere for the present day.

The referee bases part of their concerns on their inferences for the Kuroshio, while our analyses are for the Gulf Stream extension. There are some parallels in that both western boundary currents carry nutrient streams. However, there are important differences in that the Gulf Stream includes an overturning component and that current extends along a north-eastward pathway and provides a strong connection to the subpolar gyre, while the Kuroshio does not include that overturning component.

For the North Atlantic, there are a range of papers that support the view that the nutrient stream of the Gulf Stream sustains biological productivity: see papers by Pelegri and Csanady (1991), Pelegri et al. (2006), Williams et al. (2006, 2011) and Whitt and Jansen (2020). All cited in our manuscript.

Additional Supplementary Figure S3 revealing vertical profiles of the partition of nutrients into preformed and regenerated and associated transports for the Florida Current (in accord with Figure 4).

Figure S3. Biogeochemical profiles and transports in the Florida Straits at 27°N (as in Fig. 4): (a) nitrate (blue line) partitioning into preformed (orange line) and remineralised (yellow line) nutrients ($\mu\text{mol kg}^{-1}$); (b) nitrate transport integrated from the surface versus depth (kmol s^{-1}); (c) nitrate in density coordinates ($\mu\text{mol kg}^{-1}$); and (d) nitrate transport integrated from the surface versus neutral density (kmol s^{-1}); (e) DIC (blue line) and pre-industrial DIC (orange line) ($\mu\text{mol kg}^{-1}$); (f) DIC transport integrated from the surface versus depth (kmol s^{-1}); (g) DIC versus density ($\mu\text{mol kg}^{-1}$) and (h) associated transport integrated from the surface versus neutral density (kmol s^{-1}); (i) anthropogenic carbon (blue line) together with saturated anthropogenic carbon (orange line) that is the maximum possible anthropogenic carbon that can be held and potential to uptake additional anthropogenic carbon (yellow line) ($\mu\text{mol kg}^{-1}$); (j) anthropogenic carbon transport integrated from the surface versus depth (kmol s^{-1}); (k) anthropogenic carbon versus density and (l) associated transport integrated from the surface versus neutral density (kmol s^{-1}).

Thank you, we feel that addressing both your central concerns and providing a wider context has strengthened the manuscript.

Reviewer #2 (Remarks to the Author):

Please find enclosed a file with my comments to the authors.

I'm happy to sign this review, Josep L. Pelegrí.

Review of “Biogeochemical streams supply nutrients and depleted anthropogenic carbon over the North Atlantic” by R. G. Williams et al., submitted to Communications Earth & Environment

The manuscript entitled “Biogeochemical streams supply nutrients and depleted anthropogenic carbon over the North Atlantic”, by R. G. Williams et al., addresses the relevance of the Gulf Stream as a major subsurface conduit for the transport of biogeochemical properties from the subtropical to the subpolar North Atlantic, in what constitutes the last stretch of the upper or returning limb of the Atlantic Meridional Overturning Circulation (AMOC). The manuscript reviews ideas developed during the last decades on the key role of the Gulf Stream exporting heat and (inorganic) nutrients from the tropics and subtropics into the subpolar North Atlantic, and further explores more recent developments on the associated transport of dissolved inorganic carbon (DIC) in relatively old waters, depleted in anthropogenic carbon, and their potential impact on climate. (Hereafter, when saying “nutrients” I will always refer to “inorganic nutrients”.)

In my view the manuscript has two main relevant contributions:

1) It reviews historical concepts, and revisits them with several new plots, in a comprehensive way, hence providing a joint view on how the Gulf Stream transports three properties that are key for climate – heat, nutrients and DIC, specifically DIC anomalies of anthropogenic origin – into the subpolar North Atlantic.

2) It extends the relatively recent work by S. Ridge and G. McKinley (Advective controls on the North Atlantic anthropogenic carbon sink. *Glob. Biogeochem. Cycles* 34, e2019GB006457, 2020) on the supply of subsurface waters depleted in anthropogenic carbon to the subpolar North Atlantic, exploring the extra-subtropical sources.

Placing these ideas together has substantial merit and would likely grant the publishing of this manuscript.

Thank you for your positive overall comments.

However, in my view, there are a number of significant aspects related to the above main contributions that ought to be completed and/or clarified before the manuscript is published. I will try to explain them next, followed with an enumeration of additional minor issues that I believe should also be considered.

Thank you for the detailed review and the points raised.

We pleased to reference your latest review.

Pelegrí, J. L., Vallès-Casanova, I., & Orúe-Echevarría, D. (2019). The gulf nutrient stream. *Kuroshio Current: Physical, Biogeochemical, and Ecosystem Dynamics*, 23-50.

Historical concepts on heat and nutrient transport

The subsection on **Gulf Stream and heat transfer** begins referring to the Gulf Stream heat transport by citing relatively new references. Here I miss the mentioning of early seminal work on northward heat transport by Carl Wunsch and collaborators (e.g. Wunsch and Grant, 1982; Wunsch, 1984; Rintoul and Wunsch, 1991).

We agree that referencing the seminal work of Carl Wunsch is merited.

However, Wunsch and Grant (1982) is primarily focussed on developing an inverse box model and applying geostrophy to obtain the circulation, which confronts a prior view of tight recirculations by Worthington (1976). Instead Roemmich and Wunsch (1985) uses hydrography and 2 zonal sections with an inverse model to estimate a poleward heat flux of 1.2 PW across 24°N and 0.8 PW across 36°N. In addition, Rintoul and Wunsch (1991) uses hydrography and 2 zonal sections

with an inverse model including nutrient constraints to estimate a poleward heat flux of 1.3 ± 0.2 PW across 24°N and 36°N , and a northward nitrate flux of 119 ± 35 kmols⁻¹ across 36°N .

So we have now referenced Roemmich and Wunsch (1985) and Rintoul and Wunsch (1991).

In the third line of the next subsection, on **Gulf Stream and downstream nutrient transfer**, the authors state “While there is a clear sea surface temperature signal, there is little contrast in surface nutrient concentrations (Fig. 2b).”

Sorry we needed to be more explicit. There is a clear sea surface temperature contrast between the subtropical and subpolar flanks of the Gulf Stream. That contrast is not as marked for nutrients in the surface mixed layer.

Figure 2 does not show the temperature (we think that the referee means nutrients) field, so there is no way to know if there is a temperature (nutrient) contrast between the Gulf Stream and the gyre interior (I imagine this is what the authors mean), and it is difficult to appreciate if the surface nutrient contrast is significant. In my opinion, the important point here is that there is little along-isopycnal contrast in the concentration of nutrients, this should be clearly stated.

Sorry this is a misunderstanding from our prose not being explicit enough: along density surfaces, we do not think that there is a significant nutrient contrast, but there is at the sea surface or within the mixed layer.

See L64-65

While there is a clear sea surface temperature signal across the Gulf Stream, there is little contrast in surface nutrient concentrations (Fig. S1).

The authors state that the “The nutrient stream provides a nutrient communication pathway over the basin” and cite Pelegrí and Csanady (1991) and Pelegrí et al. (1996). I think this should be clarified, the basin-wide nutrient communication responds to the subtropical along-isopycnal anticyclonic basin-wide recirculation and the associated progressive remineralization of the organic matter that is subducted at higher latitudes. The outcome is the “nutrient bearing stratum (NBS)” and, in particular, the low isopycnal contrast in nutrient concentration between the core of the nutrient stream and the interior subtropical waters. The work by Pelegrí and Csanady (1991) and Pelegrí et al. (1996) focused on how this along-isopycnal basin-wide clockwise increase in nutrients is channeled northward via the Gulf Stream as a subsurface along-isopycnal stream of nutrients – which was progressively depleted as waters reached the sea surface – and pioneered the calculations on the intensity of the nutrient stream, the associated nutrient transport and its northward penetration and irrigation of the subpolar region.

Agreed, that is the above point we are referring to and cite at the outset the important work of Pelegrí and Csanady (1991) and Pelegrí et al. (1996).

I believe this is also a good opportunity for the authors to briefly refer to the original work behind the NBS, in particular work by Jorge Sarmiento and collaborators in the 1980s, as well as the review by Gabe Csanady (Physical basis of coastal productivity: The SEEP and MASAR experiments, EOS, 71, 1060-1065, 1990) where the NBS term was first coined. Subtropical versus remote water, nutrient and carbon sources

We have referenced the recent review by Pelegrí et al. (2019) that includes the NBS and provides that historical context. However, only part of our study is about the nutrient pathways and we do not wish to bring in new terminology in this *Perspective*, as we wish to make the article accessible to a wider readership. We have not previously used the nutrient bearing stratum terminology and if there is a related nutrient term, we would prefer to focus on the transport-weighted nutrient concentration as in Williams et al. (2011).

The source of the Gulf Stream waters with high concentration of nutrients (and low concentration of anthropogenic-carbon depleted carbon, see below) is, or should be, a central issue of this manuscript. The authors provide several pieces of information but I feel they fail to close the story in a convincing manner, or at least to clearly point at what we still do not know about this issue.

Our focus is on the Gulf Stream providing sub-surface horizontal fluxes of nutrients that pass into the subpolar gyre. However, the referee is correct that there are also subtropical pathways.

To provide a more complete context, we also now include lighter pathways of the nutrient stream, which reside in the subtropical gyre, rather than the denser pathways connecting to the subpolar gyre. We have expanded our figure of the modelled nutrient streams to include those lighter surfaces. Figure 3 now includes the nitrate flux and the dissolved organic nitrogen flux for potential sigma layers lighter than 26.8.

Modified figure 3:

Figure 3. Nutrient Stream over the Atlantic basin from an eddy-permitting model³⁰: (a) horizontal nitrate flux ($\text{mmol s}^{-1}\text{m}^{-2}$) integrated over density layers less than $\sigma = 26.8 \text{ kg m}^{-3}$; (b) dissolved organic nitrogen flux ($\text{mmol s}^{-1}\text{m}^{-2}$) for the same layers as in (a); and (c) nitrate flux ($\text{mmol s}^{-1}\text{m}^{-2}$) for density layers integrated over $26.8 < \sigma < 27.5 \text{ kg m}^{-3}$. The modeled nitrogen transport is from an isopycnic model MICOM with 0.23° horizontal resolution coupled with a biogeochemical model. The associated meridional transports across the basin are shown in Fig. S2. In (d), a schematic view of the nutrient stream pathways³³ (green arrows) over the Atlantic basin with gyre circulations (black lines), upwelling sites, and mode water formation regions (grey regions).

In terms of the origin of the nutrients carried in the nutrient stream, this origin varies with the temperature and density class of the waters. We are happy with our discussion of that source based on Palter and Lozier (2008) and Williams et al. (2006), reported in the manuscript.

In detail for the referee, there have been water-mass studies for the origin of the waters within Florida Straits, which have been associated with two sources; relatively fresh water dominating two temperature classes, cooler than 12°C [corresponding to denser than $\sigma=27 \text{ kg m}^{-3}$] that comes from the southern hemisphere and warmer than 24°C and is from the recirculated water from the north Atlantic subtropical gyre (associated with the salinity maximum between 12 and 24°C , Schmitz and Richardson, 1991). These two sources contribute approximately equally to the transport through the Straits with the North Atlantic recirculation contributing 17 of 30 Sv and the South Atlantic component the remaining 13Sv (Schmitz et al 1993). Szuts and Meinen (2017) updated this analysis of water masses within Florida Straits using more repeat hydrographic sections collected over a twenty-year time period and identifying water mass boundaries by density rather than temperature interfaces. The most substantive difference between this and previous

studies was that they added a fourth water mass, splitting the intermediate North Atlantic water into an eastern and western component based on the Ertel potential vorticity. The interfaces separating the surface and deepest fresh water are very close to the 12 and 24°C interfaces identified by Schmitz and Richardson (1991), and, crucially for our study, the coolest freshest water cooler than 12°C was still associated with a southern hemisphere source.

The water below 12°C, with a southern hemisphere source, is the coldest, freshest water on the section, with the highest nutrient and lowest anthropogenic carbon concentrations.

References to support the above water-mass analyses:

Schmitz Jr, W.J. and Richardson, P.L., 1991. *On the sources of the Florida Current. Deep Sea Research Part A. Oceanographic Research Papers*, 38, pp.S379-S409.

Szuts, Z.B. and Meinen, C.S., 2017. *Florida current salinity and salinity transport: Mean and decadal changes. Geophysical Research Letters*, 44(20), pp.10-495.

In our text, we have updated Figure 4 to show more property distributions in the Florida Straits, including the salinity and preformed nutrients.

New Figure 4 including isotherms 12°C and 24°C (white contours) on the salinity plot in Fig. 4a with the 12°C surface being very close to the sigma=27 surface.

Updated figure 4.

Figure 4. Biogeochemical Stream in the Florida Straits at 27°N: (a) salinity (psu) (with white contours for 12°C and 24°C); (b) nitrate concentration ($\mu\text{mol kg}^{-1}$); and (c) preformed nitrate ($\mu\text{mol kg}^{-1}$); (d) meridional velocity (m s^{-1}); (e) meridional nitrate transport (kmol s^{-1}); (f) meridional preformed nitrate transport (kmol s^{-1}); (g) dissolved inorganic carbon (DIC) ($\mu\text{mol kg}^{-1}$); (h) anthropogenic carbon ($\mu\text{mol kg}^{-1}$); (i) capacity of waters to absorb additional anthropogenic carbon (anthropogenic carbon uptake potential, $\mu\text{mol kg}^{-1}$); (j) northward DIC transport (kmol s^{-1}); (k) northward anthropogenic carbon transport (kmol s^{-1}); and (l) northward anthropogenic carbon uptake potential transport (kmol s^{-1}). The transports are calculated by combining property fields with average water velocities from a series of hydrodynamical cruises in 2012^{38,39} and the plots include neutral density contours 26.5 to 27.5 kg m^{-3} .

In the section on **Gulf Stream and downstream nutrient transfer**, the authors state that “The nutrients carried in the nutrient stream mainly originate from outside the subtropics, and are supplied from mode and intermediate waters originating from the Southern Ocean”, citing Sarmiento et al. (2004) and Palter and Lozier (2008). Indeed, these authors proved that subantarctic mode waters are the main remote source of waters arriving to the Gulf Stream waters through the Florida Straits but this is not the same to saying that the nutrient load of the Gulf Stream is dominated by nutrients preformed in the subantarctic ocean.

Our focus is on the properties carried from the Florida Current, we now include a plot of preformed nutrients in Figure 4c.

Palter and Lozier (2008) explored the source of the Gulf Stream nutrients using an oxygen-corrected phosphate tracer. They diagnosed that subtropical waters are important for supplying the nutrients in light waters ($26.0 < \sigma < 27.0 \text{ kg m}^{-3}$) accounting for 85%, but that for denser waters ($27.0 < \sigma < 27.5 \text{ kg m}^{-3}$) layers mostly originate from outside the subtropics (80% or more); they refer to those waters as of tropical origin but we expect them to ultimately originate from the subantarctic (as depicted in the schematic in Figure 4d). These denser waters are the nutrients being transported into the subpolar gyre.

The argument somehow parallels heat transport, the returning limb of the AMOC comes from the subantarctic region but we know that most heat transport by the Gulf Stream does not come from that far.

The water within Florida Straits has previously been associated with two sources; relatively fresh water dominating two temperature classes (cooler than 12°C and warmer than 24°C) that comes from the southern hemisphere and the recirculated water from the north Atlantic subtropical gyre associated with the salinity maximum between 12 and 24°C (Schmitz and Richardson, 1991). These two sources contribute approximately equally to the transport through the Straits with the North Atlantic recirculation contributing 17 of 30 Sv and the South Atlantic component the remaining 13Sv (Schmitz et al 1993)..

Indeed, several arguments show that most of the nutrient load in the nutrient stream likely has its source on the subtropics.

The work of Palter and Lozier (2008) is very clear in identifying the nutrients carried in the denser waters in the Florida Strait originate from outside the subtropics. We agree that lighter waters in the Gulf Stream acquire their nutrients from the subtropics, but the denser waters carried in the Gulf Stream (that pass into the subpolar gyre) acquire their nutrients from the tropics or ultimately from mode waters subducted in the sub Antarctic.

First, the actual upper-limb contribution of the AMOC is no more than 20 Sv so necessarily a substantial fraction of the northward flow through the Florida Straits (about 35 Sv) and east of Bahamas (less than 5 Sv) is subtropical water recirculating through the Sargasso Sea.

Agreed.

Second, between the Florida Strait and 36°N the Gulf Stream (down to 2000 m) roughly doubles its water transport but triples its nutrient transport (Pelegrí and Csanady, 1991); this additional nutrient-rich water source is essentially nutrient-rich water recirculating in the subtropical gyre.

We agree with some of the points of the referee. The above statement is only correct for the full depth water column based upon the analysis of Williams et al. (1991) (see below Table 1 from Williams et al. (1991)). In accord with the referee, there is a downstream increase in the full depth, volume transport by factors of 2.6 between Florida Strait and 35.5°N , and factor of 4.6 between Florida Strait and 36.5°N ; while there are slightly larger downstream nitrate transport increases of 3.1 between Florida Strait and 35.5°N , and factor of 6.8 between Florida Strait and 36.5°N .

However, if we focus on the crucial density layers that are supplied to the subpolar gyre then the opposite conclusion is obtained.

Focusing on the sigma layers $\sigma=27.1$ to 27.5 kg m^{-3} (that passes into the subpolar gyre):

1. We do find the downstream volume transport increases from 1.7 Sv at 27°N, 7.8 Sv at 35.5°N and 13.7 Sv at 36.5°N, representing a downstream volume transport increases by factors of 4.6 and 8.0.
2. There is also a downstream increase in nitrate transport in these layers from 49.7 kmol s⁻¹ at 27°N, 169.1 kmol s⁻¹ at 35.5°N and 306.6 kmol s⁻¹ at 36.5°N, representing downstream nutrient transport increases by factors of 3.4 and 6.2.
3. However, the factors of the downstream nutrient transport increase are *smaller* than those factors for the downstream volume transport increases.
4. Equivalently the transport-weighted nitrate concentration decreases downstream from Florida Strait to the separated Gulf Stream, such as values of 28.45 μmol kg⁻¹ at 27°N decreasing to 21.89 μmol kg⁻¹ at 36.5°N for σ=27.2 kg m⁻³ layer.

Thus, the analysis for the dense layers supplied to the subpolar gyre supports our view that the nutrients are not mainly supplied by the subtropical recirculation, but instead are supplied by the overturning circulation linked to the delivery of waters older than 12°C and denser than σ=27.0 kg m⁻³.

Table 1. Diagnostics of Volume Transport (Sv), Nitrate Transport (kmol s⁻¹) and Transport-Weighted Nitrate Concentration (μmol kg⁻¹) in Different Density Classes for the Gulf Stream at 27°N, 35.5° and 36.5°N^a

σ _θ Range	Volume Transport $\int v dx dz$ (Sv)			Nitrate Transport $\int \rho v NO_3^- dx dz$ (kmol s ⁻¹)			Transport-Weighted Nitrate $\frac{\int \rho v NO_3^- dx dz}{\int \rho v dx dz}$ (μmol kg ⁻¹)		
	27°N	35.5°N	36.5°N	27°N	35.5°N	36.5°N	27°N	35.5°N	36.5°N
σ _θ < 26.2	15.8	18.4	15.6	39.8	38.2	26.2	2.51	2.08	1.68
26.2 ≤ σ _θ < 26.5	4.3	15.4	23.6	33.8	72.2	84.2	7.94	4.69	3.57
26.5 ≤ σ _θ < 26.8	6.0	11.2	17.7	85.2	118.6	153.9	14.15	10.59	8.69
26.8 ≤ σ _θ < 27.1	4.2	9.6	12.7	94.5	162.0	217.7	22.26	16.88	17.14
27.1 ≤ σ _θ < 27.3	1.4	4.1	7.4	39.3	87.2	162.0	28.45	21.27	21.89
27.3 ≤ σ _θ < 27.5	0.3	3.7	6.3	10.4	81.9	144.6	30.50	22.14	22.95
27.5 ≤ σ _θ	0.0	20.3	66.2	0.0	379.6	1270.6	0.00	18.70	19.19
σ _θ < 27.1	30.3	54.6	69.6	253.3	391.0	482.0			
σ _θ < 27.5	32.1	62.4	83.3	303.0	560.1	788.6			
Total	32.1	82.7	149.5	303.0	939.7	2059.2			

^aUncertainties in the transports are typically 10%.

Third, the along-isopycnal continuity of nitrate concentration (Fig. 2b) and the subtropical pathways in the submitted manuscript (Figure 5) also support the idea that most of the nutrient stream has its source on the subtropics, with the highest nutrient concentrations corresponding to water recirculating through the long Sargasso-Sea subtropical pathway.

We agree that the prior version of our manuscript overlooked this aspect. To provide a more complete view, we now illustrate the subtropical pathways with new panels in Figure 3a,b,c for nitrate and DON for σ < 26.8 kg m⁻³. However, our particle trajectories from the Florida Strait show that the communication to the subpolar gyre is much more striking for deeper and denser waters passing through Florida Strait.

Fourth, the works by Williams, McDonagh, Roussenov et al.

(Nutrient streams in the North Atlantic: Advective pathways of inorganic and dissolved organic nutrients. *Global Biogeochemical Cycles*, 25, GB4008, 2011) and Holzer and Primeau (*Global teleconnections in the oceanic phosphorous cycle: Patterns, paths, and timescales. Journal of Geophysical Research Oceans*, 118, 1775-1796, 2013) show that the load of subantarctic nutrients incorporated into the Gulf Stream is relatively small.

Sorry we differ with that interpretation for the denser layers. Again Palter and Lozier (2008) show that denser waters in the Gulf Stream (27.0 < σ < 27.5 kg m⁻³) mostly originate from outside the subtropics (80% or more).

Fifth, and final, Figure 3 of the manuscript is interpreted by the authors as implying that “nutrients carried in the nutrient stream mainly originate from outside the subtropics, and are supplied from mode and intermediate waters originating from the Southern Ocean”.

In the study of Williams, Roussenov and Follows (2006), we included tracer experiments to show how water masses from the South Atlantic or Southern Ocean are transported to the Florida Current and then transported further north via the nutrient stream of the Gulf Stream.

We now include an additional schematic that we have previously used to illustrate this connection in Figure 4d from Williams, Roussenov and Follows (2006).

Methods section in the manuscript does not explain how this is done but I understand that the figure shows transports in 0.23° bins in the isopycnal layer between 26.8 and 27.5 kg m^{-3} . If this is the case, I cannot agree that the figure implies a major extra-subtropical source. The figure is difficult to interpret (there are large complex gyres, and I wonder if there is a white color in the color bar?) but I would say that the northern transport at $(25^\circ\text{N}, 80^\circ\text{W})$ and $(25^\circ\text{N}, 75^\circ\text{W})$ is very intense but also very narrow, while the western transport at 70°W is also intense but very wide, extending between 23°N and 27°N , largely feeding from the subtropical cycle. To actually see these contributions a plausible strategy would be to calculate and plot the zonally integrated values of meridional nitrate transport.

We have cut the panel that the referee questioned in Figure 3. The white colour is to avoid plotting low magnitude contributions.

We agree that it is difficult to interpret the zonally-integrated effects so have included in the supplementary figure S2 side bars to show the zonally-integrated volume and nitrate transport. Within the σ range of 26.8 to 27.5 kg m^{-3} , there is a northward volume transport (blue dashed line) and a northward nitrate transport (black full line).

Additional supplementary figure S2 including side panels for transports.

Figure S2. Nutrient Stream from an eddy-permitting model³⁰ (as in Fig. 3): (a) the horizontal nitrate flux ($\text{mmol s}^{-1} \text{m}^{-2}$) integrated over density layers less than $\sigma = 26.8$; (b) the dissolved organic nitrogen flux ($\text{mmol s}^{-1} \text{m}^{-2}$) for the same layers as in (a); and (c) the horizontal nitrate flux ($\text{mmol s}^{-1} \text{m}^{-2}$) for density layers integrated over $26.8 < \sigma < 27.5$. In the right side panels, there is the meridional volume transport over the width of the basin for those layers (Sv, blue line) and the associated nutrient transport (kmol s^{-1} , black line).

Our crucial point again is the delivery of the bottom waters in the Florida Strait sections in Figure 4 includes low anthropogenic carbon and elevated nutrients that are in accord with sub-Antarctic mode water sources.

I would like to also suggest the authors to look at the review article by Pelegrí et al. (The Gulf

Nutrient Stream. In *Kuroshio Current: Physical, Biogeochemical and Ecosystem Dynamics*, Eds. T. Nagai, H. Saito, K. Suzuki, M. Takahashi. AGU-Wiley, GMS 243, 23-50, ISBN 9781119428343, 2019), as it is not cited in the manuscript despite it contains an exhaustive revision of many of the above ideas.

Thank you. We are happy to include this reference.

A source of subsurface waters depleted in anthropogenic carbon

A good understanding of the partition between subtropical- and remote-source contributions is crucial when we look at the Gulf Stream transport of subsurface waters depleted in anthropogenic carbon (Cant). In my opinion, here it is very important to consider the time scales of the different pathways, as these define their “age”, i.e. their Cant at origin and hence their uptake capacity. Waters that are part of the upper (returning) limb of the AMOC and were subducted in the subantarctic region will take many years to reach the Gulf Stream (as they may recirculate in the subtropical and tropical oceans several times), while waters subducting in the North Atlantic will take much less to reach the Gulf Stream.

We agree that this distinction is important and this separation of pathways and timescales is defined by density class. The waters subducted in the subtropical North Atlantic are in lighter density classes than the maximum winter density outcrops of $\sigma=27 \text{ kg m}^{-3}$. For denser waters, $\sigma>27 \text{ kg m}^{-3}$, those waters have mainly originated from outside the subtropics (as discussed above), so that these waters have much higher ages consistent with the longer pathways from the Southern Ocean.

For simplicity, suppose these times are 50 and 10 years, respectively, so to assess today's absorbing capacity of these different waters the relevant figures would be the CO₂ atmospheric concentrations in 1974, 2014 and 2024, which respectively are 330, 399 and 425 ppm. The actual absorbing capacity of the subsurface Gulf Stream waters reaching the subpolar ocean will hence depend on a proper assessment of the travel times and relative contributions of the different water masses.

Evaluating the relative importance of the different sources goes beyond the scope of this study. However, we can make a rough calculation based on the estimates of the referee. As stated before, Palter and Lozier (2008) identified that the nutrients in the denser parts of the Gulf Stream ($27.0 < \sigma < 27.5 \text{ kg m}^{-3}$) predominately originate from outside the subtropics (80% or more). So based on these estimates the signal would be 80% x 330 ppm for the Southern Ocean source and 20% x 399 ppm for the subtropical source, which gives a cumulative signal of 343 ppm, which is much less than the present day value of 425 ppm and close to the 330 ppm value for 1974.

In the section **Sensitivity of subpolar carbon to upstream waters** the manuscript explores the extension of the upstream region that influences the carbon content in the subpolar North Atlantic at 4 and 8 years (or something alike, I would appreciate a better description of what the authors mean by “sensitivity”, I cannot either see a proper explanation in the **Methods**). These are relatively short times (it would be nice to see the results for longer times) but nevertheless we appreciate an upstream influence into both the Caribbean and the northwestern portion of the gyre, in what would constitute a smaller recirculation region of the subtropical gyre (see Figure 9 in Pelegrí et al., 2019). To me this suggests that there is a subtropical source (as proposed by Ridge and McKinley, 2020) of relatively young waters, so with relatively little uptake capacity, as well as a remote source of much older waters, largely corresponding to the upper limb of the AMOC.

We thank the reviewer for this comment.

We have clarified the meaning of “sensitivity” in the Methods section. Here, sensitivity is explicitly defined as the change in the annual mean dissolved inorganic carbon (DIC) content, averaged over a selected “control volume” (the upper 500 m of the subpolar North Atlantic, as depicted by a red dashed line in Figure 7), in response to hypothetical 14-day scale anomalies in upstream DIC concentrations. It is essential to note that altering the timescale of either the objective function (e.g., averaging over a different period) or these hypothetical perturbations would impact the interpretation of the resulting sensitivities.

Regarding the suggestion for longer time scales, our present adjoint analysis focuses on lead times of 1, 4, and 8 years. On longer timescales, the linearity of the response to typical perturbations (of the magnitude that one might expect given the observed variability) may start to break down, and the adjoint approach is limited to providing linear sensitivities. Nevertheless, these timescales effectively demonstrate the upstream influences we wish to highlight.

We appreciate the reviewer's insightful interpretation of our results regarding the different water masses and their implied age and uptake capacities. While the mean state and long-term carbon uptake are influenced by the properties of these various water masses, our adjoint sensitivity analysis specifically demonstrates how *anomalies* (perturbations) in upstream DIC concentrations within the Gulf Stream and its extension directly affect the carbon content in the subpolar North Atlantic. This explicitly reveals the dynamic connectivity and influence of upstream processes on the subpolar carbon sink, independent of the complexities of the mean water mass origins. Our adjoint results indeed highlight the Florida Straits and Gulf Stream as prominent regions where DIC perturbations can affect the objective function, and our analysis shows how this signal propagates along the Gulf Stream into the subpolar North Atlantic.

Add the text to the Methods L241-249

The sensitivity of the carbon content is defined as the change in the annual DIC content, averaged over a selected "control volume" (the upper 500 m of the subpolar North Atlantic, as depicted by a red dashed line in Figure 7), in response to hypothetical 14-day scale anomalies in upstream DIC concentrations. The sensitivity fields are scaled and then normalized for each time slice. Firstly, the raw sensitivity, $\partial J/\partial \text{DIC}$, is scaled by multiplying by the spatial standard deviation, σ_{DIC} , of the DIC and then divided by the cost function, J , such that

$$S_J = (\partial J/\partial \text{DIC}) \sigma_{\text{DIC}}/J$$

Secondly, the scaled sensitivity, S_J , is normalized by dividing by its maximum value on that isopycnal from lag year 1, allowing the sensitivities to be compared across different times. The sensitivity contains some small negative values that have a magnitude of only 1% of the maximum positive value so that we focus on the positive values for S_J . The sensitivities are mapped in space and time for a density layer centered on $\sigma = 27.5 \text{ kg m}^{-3}$ to reveal the upstream control of the subpolar DIC content.

Modified text L151-158:

The adjoint model provides the sensitivity of that carbon content to upstream carbon carried in a potential density layer centered on $\sigma_\theta = 27.5 \text{ kg m}^{-3}$. For a lead time of 1 year, the subpolar carbon content is only sensitive to carbon held within its subpolar domain (Fig. 7a). For a lead time of 4 years, there is a band of high sensitivity running along the path of the Gulf Stream and the western side of the subtropical gyre and, for a lead time of 8 years, that high sensitivity band extends from the Gulf Stream to the southwestern side of the subtropics (Fig. 7b,c). Hence, the adjoint sensitivity maps reveal upstream influence from the Florida Straits and the Gulf Stream extension into the subpolar North Atlantic on lead times of 4 to 8 years. This analysis suggests that the carbon content of the subpolar gyre is influenced by waters originating from the subtropical gyre.

Additional minor issues

- In the **Introduction**, the authors refer to the Gulf Stream as a "biogeochemical stream" that "provides a sub-surface redistribution of nutrients and depleted anthropogenic carbon... which extends over the North Atlantic". Two issues appear here. First, to my knowledge, this is the first time the "biogeochemical stream" terminology is used, so it should be stated clearly. Second, the term "biogeochemical stream" is, in my opinion, incomplete as "biogeochemical" is an adjective (such as "geochemical", Oxford Dictionary), so "biogeochemical streams" refer to streams that have some "biogeochemical" characteristics but without stating which ones. This contrasts with "nutrient streams", as "nutrient" is both an adjective and a noun (Oxford Dictionary), so "nutrient stream" becomes a compound noun, a stream that has or carries nutrients.

We wish to refer to how the Gulf Stream acts as a biogeochemical stream providing a redistribution of dissolved inorganic nutrients and depleted anthropogenic carbon, which we refer to in this study.

We have already shown via our field work and modelling that this viewpoint carries over to dissolved organic nutrients (Williams et al., 2011, GBC, doi:10.1029/2010GB003853); included now in a revised version of Figure 3. We do not see any reason why the idea should not carry over for other biogeochemical variables, such as an elevation in other macro-nutrients and a likely depletion in trace metals.

We agree that biogeochemical is being used as an adjective, but we do not see why that is a problem. We are happy to take advice from the Editor.

- The **Introduction** ends up saying “future possible changes in carbon uptake from the atmosphere are illustrated for the North Atlantic that are separated into the ocean carbon response and feedback due to rising atmospheric CO₂ plus the ocean carbon feedback due to climate change”. The two concepts are intimately related so, to avoid confusion, I suggest that you clarify what you mean by “ocean carbon feedback due to climate change”.

There is a large body of work that defines this separation of carbon feedbacks due to atmospheric CO₂ versus carbon feedbacks due to climate change; see Arora, V.K et al., 2020. Carbon-concentration and carbon-climate feedbacks in CMIP6 models and their comparison to CMIP5 models. *Biogeosciences*, 17, 4173-4222, doi:10.5194/bg-2019-473 and references contained in there.

The climate change effect includes changes in temperature, stratification and circulation.

- Line 5 of subsection **Gulf Stream and heat transfer** reads “there is a maximum in heat loss to the atmosphere from turbulent heat fluxes”; I believe this should read “there is a maximum in heat loss to the atmosphere from air-to-sea turbulent heat fluxes”.

L47 Changed to “there is a maximum in the net heat loss to the atmosphere”

- I believe that the last two sentences of this same paragraph (“The Gulf Stream makes a crucial... taken up by the atmosphere by 50°N”) contain information related to the beginning of this same paragraph, so they appear somewhat misplaced.

Rephrased L50 to L55

The Gulf Stream makes a crucial contribution to ocean heat transport by being part of the upper limb of the meridional overturning circulation, which carries warm water northward and colder deep water southward. This heat transfer is important in warming the atmosphere with typically half of the ocean heat carried by the meridional overturning circulation at 25°N being taken up by the atmosphere by 50°N. This ocean transport and release of heat to the atmosphere, augments the seasonal heat release from the ocean to the atmosphere, and together contributes to the mild winter climate of western Europe.

- The second paragraph of this subsection (starting with “There is an ongoing debate...”) is, in my opinion, confusing. I would suggest rewriting and clarifying the ideas in this paragraph.

Cut that previous paragraph and rephrased as above.

- In section **Connectivity of the Gulf Stream and the subpolar gyre** the manuscript refers to particles released in the Florida Strait at several depth ranges. I think that, more than a depth range, the relevant issue is the isopycnal range, as the water penetration of the Gulf Stream extension into the subpolar gyre is largely set by the outcropping of isopycnals, and in the Florida Straits the isopycnals are very tilted with e.g. the 27.0 isopycnal ranging between 100 and 550 m. In my opinion, this creates uncertainties in the interpretation of the results.

The experiments were performed with an ocean model (ECCO4) that has a vertical coordinate in terms of depth rather than density, and a resolution of one degree. It is then difficult to organise the particle seeding in terms of density and match the exact isopycnal slopes found in highly resolved observations across Florida Strait. While in principle we agree that there could be an ambiguity due to the tilt of the density surfaces, the actual model results appear very clear and show the propagation of particles at depth reaching the high latitudes.

- In this same section, at the end of page 4 and beginning of page 5, the manuscript reads “Sub-surface waters have high nitrate concentrations in the subtropics and there is a

downstream surface increase in the nitrate as density layers shoal and outcrop in the subpolar gyre (Fig. 6a).” I believe this is not correct: the extension of the nutrient stream into the subpolar gyre is indeed accompanied by a wedge of nutrients, what Pelegri et al. (2006) called an irrigation of nutrients, but the nutrient concentration does decrease along this pathway.

We have rephrased the text and that point is not now there. We agree that the nutrients decrease in concentration along a density layer. See below scatter plot for nitrate data plotted in the same way as carbon data (plotted in Figures 6a and S4a) with downstream nutrient concentrations in a neutral density layer 27 to 27.5 kg m⁻³ versus downstream distance.

Sorry our prose should have been more explicit, our point instead is that there is a downstream increase in surface nutrient concentrations. See the maps in Supplementary figure S1.

In addition, our analysis of historical data revealed a downstream increase in surface nutrient concentrations, also see Fig. 3c,d for nitrate and dynamic height in Williams et al. (2006).

- Figure 6: The distribution of properties along the selected pathway is difficult to appreciate. We have updated the plots for Figure 6 to include neutral density surfaces and included accompanying scatter plots for a particular density layer in the supplementary Figure S4.

Figure 6. Biogeochemical properties along a possible Gulf Stream pathway extending from the subtropical to the subpolar gyre for (a) dissolved inorganic carbon (DIC) ($\mu\text{mol kg}^{-1}$), (b) anthropogenic carbon ($\mu\text{mol kg}^{-1}$) and (c) anthropogenic carbon uptake potential ($\mu\text{mol kg}^{-1}$) from GLODAPv2 climatology⁴⁵. The associated air-sea difference in partial pressure of dissolved carbon dioxide ($\Delta p\text{CO}_2$, μatm) linked to the DIC distribution is shown for year 2002 in (d) together with (e) the $\Delta p\text{CO}_2$ distribution if the anthropogenic carbon is fully saturated with the atmosphere and (f) the difference in the $\Delta p\text{CO}_2$ between (d) and (e), representing the effect of the under saturation in anthropogenic carbon in (c), where the atmospheric $p\text{CO}_2$ for 2002 is 373.1 μatm . This pathway is representative of a deeper trajectory as in Fig. 5c. The sections include neutral density surfaces for 26.5, 27.0 and 27.5 kg m^{-3} .

Figure S4. Scatterplots of biogeochemical properties along neutral density layers, σ_n for 27.0 to 27.5, versus latitude along a possible Gulf Stream pathway extending from the subtropical gyre into the subpolar gyre (as in Fig. 6) for (a) DIC (blue dots) and pre-industrial DIC (red triangles) ($\mu\text{mol kg}^{-1}$); (b) anthropogenic carbon (red triangles), saturated anthropogenic carbon (blue dots) and capacity to hold additional anthropogenic carbon (yellow crosses) (all in $\mu\text{mol kg}^{-1}$, blue line); and (c) partial pressure of dissolved carbon dioxide ($p\text{CO}_2$, μatm) together with the atmospheric partial pressure for years 2002 and 2024 (black dashed and red dashed horizontal lines respectively); from GLODAPv2 climatology⁴⁵.

- In the section **Sensitivity of subpolar carbon to upstream waters** the manuscript mentions an “adjoint model” and the reader is referred to the Methods section, but this section does not explain why calling it “adjoint”. I had to go to the original reference and spend some time there to get an idea of what is meant by adjoint, I would suggest to include a short explanation to facilitate the reading of the manuscript.

We agree with the reviewer that a brief explanation of the adjoint model would be beneficial for the reader, particularly regarding the terminology.

Revised explanation in the Methods section: L215-231

An adjoint model is used to provide an efficient means of calculating the sensitivity of a chosen model output (the 'objective function') to all input parameters or initial conditions. This approach applies an adjoint operation to the numerical model, effectively converting a 'forward' set of numerical integration steps into a set of 'backward' steps, enabling explicit calculations of how model states depend on earlier steps in the numerical integration. This procedure enables the computation of sensitivities for numerous inputs in a single backwards integration, tracing the influence of upstream properties, surface forcings, and even mixing coefficients on a selected objective function. This calculation is particularly useful for comprehensive sensitivity analyses, as it can be mathematically equivalent to hundreds or even thousands of individual perturbation experiments.

A few lines later the manuscript talks about "carbon carried in a potential density layer of $\sigma_\theta = 27.5 \text{ kg/m}^3$." This is an inconsistent sentence: a layer is, by definition, a range, and a density surface (such as the 27.5 kg/m^3) has infinitesimal thickness and hence cannot carry properties. Please consider rewriting.

L249 Modified to a layer centred on $\sigma = 27.5 \text{ kg m}^{-3}$.

- Figure 7, caption: the variable J is introduced but not defined, the reader only finds out its definition after ending **Methods**.

Modified figure 7 and caption to avoid the variable J:

Sensitivity of the ocean carbon storage for the subpolar North Atlantic. The maps show the normalised sensitivity (Methods) of the volume-averaged, annual mean dissolved inorganic carbon (DIC) concentration in the upper 500 m of the subpolar North Atlantic Ocean (red dashed lines) to upstream DIC anomalies within a potential density layer centered on $\sigma = 27.5 \text{ kg m}^{-3}$ across the entire North Atlantic. Each panel depicts different lead times between the DIC anomaly and the target year of the average: (a) approximately lead time of 1 year, (b) lead time of 4 years and (c) lead time of 8 years. To highlight relative sensitivities, the fields have been scaled by their maximum value.

Figure 7. Sensitivity of the ocean carbon storage for the subpolar North Atlantic. The maps show the normalised sensitivity (Methods) of the volume-averaged, annual mean dissolved inorganic carbon (DIC) concentration in the upper 500 m of the subpolar North Atlantic Ocean (red dashed lines) to upstream DIC anomalies within a potential density layer centered on $\sigma = 27.5 \text{ kg m}^{-3}$ across the entire North Atlantic. Each panel depicts different lead times between the DIC anomaly and the target year of the average: (a) approximately 1 year, (b) 4 years and (c) 8 years earlier. To highlight relative sensitivities, the fields have been scaled by their maximum value.

- The second paragraph of **Methods** reads "carbon uptake potential ($\mu\text{mol kg}^{-1}$), is estimated as the difference between dissolved inorganic carbon concentration calculated using in situ alkalinity, salinity, temperature, silicate and phosphate data and pCO_2 of $280 \mu\text{atm}$ (preindustrial) or $369 \mu\text{atm}$ (1998 level) using CO2SYS52 and the estimated in situ anthropogenic carbon concentration in 1998." This is confusing, please clarify why "280 μatm or 369 μatm ", and why the anthropogenic rather than the total carbon concentration in 1998.

We need to use the atmospheric pCO₂ value appropriate for the time of the data collected. We had used 1998 data that corresponds to 369 μatm, but now use 2002 compilation of data that corresponds to 373.1 μatm. The atmospheric value for the pre industrial was 280 μatm.

L210-211: *in situ alkalinity, salinity, temperature, silicate and phosphate data and pCO₂ of 280 μatm (for the time of the preindustrial) or 373.1 μatm (for the time of the data collected in 2002)*

- It is possibly worth mentioning that the nutrient stream concept has also been widely used for the Kuroshio Current, see the book “Kuroshio Current: Physical, Biogeochemical and Ecosystem Dynamics”, edited by T. Nagai, H. Saito, K. Suzuki, M. Takahashi, AGU-Wiley, GMS 243, 23-50, ISBN 9781119428343, 2019.

Thank you, added this reference.

L191-192 *Biogeochemical streams should carry over for other ocean basins; for example, see the review of the biogeochemical effects of the nutrient stream for the Kuroshio in the North Pacific*⁵⁵

This is why, in their review article, Pelegrí et al. (2019) decided to use the nomenclature Gulf Nutrient Stream, which can hence be capitalized.

- Possibly the authors should also clarify that, except when indicated, nutrient streams always refer to streams of inorganic nutrients.

We have identified nutrient streams carry over to dissolved organic nutrients in our field work and modelling (Williams et al., 2011, GBC, doi:10.1029/2010GB003853).

Recommendation

I admire and worthy very much the past and present work done by the authors on the origin, fate, intensity and biogeochemical processes in the nutrient streams, and I think this is a timely review that evidences the high relevance of biogeochemical processes in the North Atlantic nutrient stream, and further explores the role of this stream related to climate change.

Thank you for your support and constructive comments.

However, I think that the manuscript will greatly benefit by a revision that (a) solves the issues raised above, so as to facilitate its reading, and (b) clarifies what we know and what we yet do not know on the origin of those waters with high concentration of nutrients and low concentration of anthropogenic-carbon depleted carbon that feed the Gulf Nutrient Stream.

Clarifying these different origins is crucial for establishing the CO₂ uptake capacity of the nutrient stream subsurface waters and its potential role controlling climate.

We agree with many of the points of the referee and are happy to have included further context via expanded figures. We only differ on some detailed points and have provided our rationale based upon our diagnostics of observations and modelling studies.

I'm happy to sign this review, Josep L. Pelegrí.

Thank you for your support for the article. We feel addressing your concerns has strengthened the manuscript.

Reviewer #3 (Remarks to the Author):

The manuscript is well written, of broad interest, and while the main thrust of the article, that the Gulf Stream is a "biogeochemical stream" already exists in various forms in the literature (as shown in the introductory review here), in this reviewer's opinion the role of the GS as a biogeochemical stream is not well understood or properly emphasized in the literature and is well deserving of being reviewed and highlighted.

Thank you for your support for the article and insightful comments.

Some specific comments:

The density layer where you show the highest nitrate flux (26.5-27.5) does appear to outcrop intermittently well south of the subpolar gyre. This has been shown in some previous studies such as <https://doi.org/10.1029/2023JC020526> and <https://doi.org/10.1002/2015GL067152>. It may be worth including something about this to note if it is minor compared to your overall points, or another factor in the full biogeochemical portfolio of the Gulf Stream, and a factor that very likely would be missing in coarser resolution modeling.

The $\sigma=27$ outcrop at the end of winter is often viewed as the subtropical-subpolar boundary, so that the $\sigma=26.5$ surface outcrops in the subtropics and $\sigma=27.5$ outcrops in the subpolar gyre. We have now included in Figure 3a,b the modelled high nitrate and DON flux for lighter density layers, $\sigma < 26.8 \text{ kg m}^{-3}$.

The nutrient stream as shown in the data from the elevated horizontal flux is partitioned then between both the subtropics and subpolar gyre. We agree that there may be intermittent variations in those outcrops and that is likely to vary interannually. Williams et al. (2011) show from cruise data in 2005 gathered at the separation of the Gulf Stream that the nitrate flux into the subpolar gyre is much larger than that remaining in the subtropical gyre.

Thank you for the recommended references. We agree with this point and have included a short paragraph in the Discussion.

L185-188 Our analysis of the role of the Gulf Stream and its extension has focussed on its basin-scale connections over the North Atlantic. There may though be important smaller scale effects, such as involving sub-mesoscale exchanges^{52, 53} and mesoscale eddy stirring⁵⁴, that affects the downstream dilution of the biogeochemical properties carried by the Gulf Stream and its extension over the North Atlantic (as suggested in Figs. 6a, S4a).

Related, in this line "This outcropping then transfers the nutrients into the winter mixed layer, ultimately helping to sustain high-latitude biological productivity." It is not clear to that this nutrient strata is only outcropped into the winter mixed layer, is it too deep in the summer to have sufficient light for photosynthesis? It seems likely that it may have sufficient light for growth in some regions even at depth.

We agree that this neglect of nitrate utilisation below the winter mixed layer is an approximation. However, the light intensity is very low below typical winter mixed layer depths of 200 m or more in the subpolar gyre, so that we think this approximation is fine.

In a more general sense, there is distinction between whether the focus is the air-sea CO₂ flux or the nutrients in the winter mixed layer. It should not ultimately matter whether the nutrient utilisation takes place in the seasonal thermocline or the mixed layer for the annual air-sea CO₂ flux. If the nutrient utilisation occurs in waters below the summer mixed layer, there initially will be no response in the air-sea CO₂ flux during summer, but there will be a delayed effect on the air-sea

CO₂ uptake in winter when those seasonal thermocline waters are entrained into the winter mixed layer.

Can you provide a sentence or two of background as to why it does/doesn't shoal more consistently or whether there is enough light for this layer to grow during the summer?

I mention this because there does seem to be some evidence that it might shoal intermittently all year, but the total BGC impact of this ephemeral shoaling may be negligible compared to the injection of nutrients into the winter mixed layer.

Our focus is the nutrient transfer from the Gulf Stream to the subpolar gyre and for this basin-scale pathway the end of winter mixed layers are typically more than 200 m deep.

We are conducting separate examinations of BioArgo floats and feel the points raised are relevant for that study. However, for our basin-scale focus, we are unsure how to address the issue of intermittent shoaling, which might be very variable and differ between subtropical and subpolar latitudes. Again this concern does not affect the annual air-sea CO₂ flux.

Figure 5 and the particle tracking experiments are compelling for the argument presented here and are nicely presented. Are these supported by any observational work and can you add some discussion on this?

We are conducting a field programme C-Streams that is including analysis of Bio-Argo floats and their evolution, as well as time series of biogeochemical properties carried by the Gulf Stream for nearly 2 years. However, the outcomes of that field work and its analysis will come out over the next year or so. This perspective article is to set out our viewpoint, as well as provide the wider context for that field programme.

It should also be noted that the ECCO4 state estimate being used to drive the particle tracking calculation is an observational estimate that includes virtually all of the data sets that can best constrain the climatology of basin-scale ocean transports (e.g. Argo, altimetry, atmospheric reanalysis data for the surface forcing).

The text on the figures needs to be larger, it is challenging to read, particularly Fig 4 I cannot read the text even when zooming in due to low resolution and small font.

Agreed, enlarged fonts on Figures 1, 4 and 5.

The line "Hence, the carbon content of the subpolar North Atlantic is sensitive to the advection of carbon anomalies from the Gulf Stream and its extension" is possibly too certain given the one experiment described and lack of observational evidence. While the modeling is very interesting and I am not too skeptical of your result, it may be more appropriate to say "This modeling suggests the carbon content of the subpolar North Atlantic is sensitive to the advection of carbon anomalies from the Gulf Stream and its extension..."

We appreciate the reviewer's careful consideration of our phrasing.

We respectfully disagree with the suggestion that our modelling inferences are based on a single experiment and, therefore, are too certain. The adjoint approach employed in this study, which identifies the sensitivity of the carbon state, is mathematically equivalent to performing many hundreds or even thousands of individual perturbation experiments to ascertain the sensitivity to forcing functions and upstream boundary conditions. This comprehensive sensitivity analysis for the carbon system has not been previously identified in the literature. Furthermore, our findings are consistent with separate adjoint studies concerning subpolar heat content by Jones et al. (2018), which similarly demonstrate that subsurface temperatures within the Gulf Stream play a considerable role in determining Labrador Sea heat content over a timescale of 5 to 10 years. Given the robust nature of the adjoint methodology in exploring system sensitivities, the original phrasing accurately reflects the strong evidence provided by our modeling approach.

L151-155 Hence, the adjoint sensitivity maps reveal upstream influence from the Florida Straits and the Gulf Stream extension into the subpolar North Atlantic on lead times of 4 to 8 years. This analysis suggests that the carbon content of the subpolar gyre is influenced by waters originating from the subtropical gyre. These sensitivity maps are also consistent with a similar analysis of how Labrador Sea heat content is sensitive to the upstream sub-surface temperature carried by the Gulf Stream on a decadal timescale.

L225-231 in Methods

An adjoint model is used to provide an efficient means of calculating the sensitivity of a chosen model output (the 'objective function') to all input parameters or initial conditions. This approach applies an adjoint operation to the numerical model, effectively converting a 'forward' set of numerical integration steps into a set of 'backward' steps, enabling explicit calculations of how model states depend on earlier steps in the numerical integration. This procedure enables the computation of sensitivities for numerous inputs in a single backwards integration, tracing the influence of upstream properties, surface forcings, and even mixing coefficients on a selected objective function. This calculation is particularly useful for comprehensive sensitivity analyses, as it can be mathematically equivalent to hundreds or even thousands of individual perturbation experiments.

The first two paragraphs of the "Discussion" section should be re-written for clarity. First you note that natural ocean carbon sinks are projected to diminish, but then say that the ocean carbon uptake is projected to increase.

Reordered the first paragraph L162-169

Climate projections suggest that the effectiveness of natural ocean carbon sinks in curbing the rise of atmospheric CO₂ is likely to diminish in the future¹, based upon an assessment of global ocean carbon-cycle feedbacks². The global ocean response to rising atmospheric CO₂ involves competing effects from (i) the rise in atmospheric CO₂ altering ocean chemistry and an ocean acidity feedback, and (ii) a smaller and opposing climate change effect, including the temperature-solubility feedback and effects of stratification and circulation changes. For the ocean carbon-cycle feedbacks evaluated over the North Atlantic, maps for the increase in ocean carbon storage⁴⁶ reveal local maxima over the Gulf Stream and much of the subpolar gyre (Fig. 8a), which are primarily due to the direct effect of the rising atmospheric CO₂ (Fig. 8b). The effect of the climate change leads to reduced ocean carbon uptake over most of the subpolar North Atlantic and Norwegian Sea (Fig. 8c)

L157 The effectiveness of natural ocean carbon sinks in curbing the rise of atmospheric CO₂ is projected to diminish in future.

This statement is widely reported as a key outcome of the last IPCC report, so prefer to retain this phrasing.

By "natural" do you mean biological? I don't think so, or do you mean that their efficiency will decrease? but not the total amount of carbon taken up because of increasing atmospheric pCO₂? I may be missing a nuance, but please clarify this a bit.

The natural carbon sink includes both the physical solubility pump and the biological pump.

In the second to last line of the manuscript "Biogeochemical streams should carry over for other ocean basins..." it might be helpful to readers to be a bit more explicit when discussing the generalizability of your ideas on the Gulf Stream as a BGC stream. This could be a useful reference in pointing to the Kuroshio: <https://doi.org/10.1002/9781119428428.ch6>.

Thank you, we agree, and now reference the substantial Kuroshio review.

Nagai, T., Saito, H., Suzuki, K. & Takahashi, M. Kuroshio current: Physical, biogeochemical, and ecosystem dynamics (John Wiley & Sons, 2019).

L191-194.

Biogeochemical streams should carry over for other ocean basins; for example, see the review of the biogeochemical effects of the nutrient stream for the Kuroshio in the North Pacific⁵⁵. The

biogeochemical streams for the other basins all participate in longer overturning pathways as part of the global overturning circulation⁵⁶, so are likely to have higher values of remineralised nutrients and dissolved inorganic carbon.

Overall a well written, accessible manuscript that I expect will be of interest to a broad audience. Thank you for your constructive comments and your positive endorsement.

Reviewer #4 (Remarks to the Author):

Previous studies that focus on the effects of increased atmospheric CO₂ and warming of ocean surface, reducing carbon uptake potential by the ocean, are ignoring the importance of western boundary currents, such as Gulf Stream, that carry the potential of anthropogenic carbon uptake in its subsurface layers toward downstream regions where these waters are outcropped.

A lack of pathway from subtropical to subpolar gyre near the surface has been remarked in other publications due to Ekman processes. However, this study presents the trajectories of Lagrangian particles released at different depth levels, showing the importance of a biogeochemical stream that extends from subtropical gyre to subpolar gyre in denser layers. The further analysis showed that subpolar DIC is largely controlled by the upstream DIC through the Gulf Stream. This subsurface transport of high carbon uptake potential by the Gulf Stream, which has not been highlighted, is very important for better predicting climate change. Also, the effect of the biogeochemical stream is not limited to local region, but it could provide large impacts at global scale since the subpolar North Atlantic is the major CO₂ sink for the atmosphere. Therefore, in my opinion, the paper can be a nice contribution to the journal, Communications, Earth and Environments, after some revisions for several points as follows.

Thank you for your positive comments.

Major Comments

- Please add the line numbers.

Apologies for their prior omission. Line numbers are now added.

- My focus is not biogeochemical process, yet it is very nice to know that the Gulf Stream is controlling at some level the subpolar North Atlantic DIC.

Thank you.

Does this also mean that the high DIC transported in the subsurface Gulf Stream is supplying negative total carbon uptake potential downstream as opposed to the anthropogenic carbon uptake potential?

Yes, that is correct: the subsurface Gulf Stream transport provides high DIC as well as both enhanced potential to uptake anthropogenic carbon. We make this point more explicit in revised Figure 6 and a new supplementary figure S4.

Figure 6. Biogeochemical properties along a possible Gulf Stream pathway extending from the subtropical to the subpolar gyre for (a) dissolved inorganic carbon (DIC) ($\mu\text{mol kg}^{-1}$), (b) anthropogenic carbon ($\mu\text{mol kg}^{-1}$) and (c) anthropogenic carbon uptake potential ($\mu\text{mol kg}^{-1}$) from GLODAPv2 climatology⁴⁵. The associated air-sea difference in partial pressure of dissolved carbon dioxide ($\Delta p\text{CO}_2$, μatm) linked to the DIC distribution is shown for year 2002 in (d) together with (e) the $\Delta p\text{CO}_2$ distribution if the anthropogenic carbon is fully saturated with the atmosphere and (f) the difference in the $\Delta p\text{CO}_2$ between (d) and (e), representing the effect of the under saturation in anthropogenic carbon in (c), where the atmospheric $p\text{CO}_2$ for 2002 is $373.1 \mu\text{atm}$. This pathway is representative of a deeper trajectory as in Fig. 5c. The sections include neutral density surfaces for 26.5 , 27.0 and 27.5 kg m^{-3} .

Figure S4. Scatterplots of biogeochemical properties along neutral density layers, γ_n for 27.0 to 27.5 , versus latitude along a possible Gulf Stream pathway extending from the subtropical gyre into the subpolar gyre (as in Fig. 6) for (a) DIC (blue dots) and pre-industrial DIC (red triangles) ($\mu\text{mol kg}^{-1}$); (b) anthropogenic carbon (red triangles), saturated anthropogenic carbon (blue dots) and capacity to hold additional anthropogenic carbon (yellow crosses) (all in $\mu\text{mol kg}^{-1}$, blue line); and (c) partial pressure of dissolved carbon dioxide ($p\text{CO}_2$, μatm) together with the atmospheric partial pressure for years 2002 and 2024 (black dashed and red dashed horizontal lines respectively); from GLODAPv2 climatology⁴⁵.

Please clarify how the carbon uptake potential provided by the Gulf Stream works in net carbon uptake, not only limited to the anthropogenic part. This is very important information for readers outside the scope of biogeochemistry.

We apologise for not being explicit enough and agree that this distinction is important.

Firstly, we have set out the partitioning of dissolved inorganic carbon into preindustrial and anthropogenic parts together with the anthropogenic carbon being compared with the saturated anthropogenic and the uptake potential for anthropogenic carbon. We show the anthropogenic carbon uptake potential in Figures 4 and 6. The supplementary figure S3 is included to show the carbon partition (see panels e and i).

Figure S3. Profiles of biogeochemical profiles and transports in the Florida Straits at 27°N (as in Fig. 4): (a) nitrate (blue line) partitioning into preformed (orange line) and remineralised (yellow line) nutrients ($\mu\text{mol kg}^{-1}$); (b) nitrate transport integrated from the surface versus depth (kmol s^{-1}); (c) nitrate in density coordinates ($\mu\text{mol kg}^{-1}$); and (d) nitrate transport integrated from the surface versus neutral density (kmol s^{-1}); (e) DIC (blue line) and pre-industrial DIC (orange line) ($\mu\text{mol kg}^{-1}$); (f) DIC transport integrated from the surface versus depth (kmol s^{-1}); (g) DIC versus density ($\mu\text{mol kg}^{-1}$) and (h) associated transport integrated from the surface versus neutral density (kmol s^{-1}); (i) anthropogenic carbon (blue line) together with saturated anthropogenic carbon (orange line) that is the maximum possible anthropogenic carbon that can be held and potential to uptake additional anthropogenic carbon (yellow line) ($\mu\text{mol kg}^{-1}$); (j) anthropogenic carbon transport integrated from the surface versus depth (kmol s^{-1}); (k) anthropogenic carbon versus density and (l) associated transport integrated from the surface versus neutral density (kmol s^{-1}).

Secondly, we illustrate the downstream dilution of the DIC within a density layer in Figures 6a and S4a, prior to that density layer outcropping in the subpolar gyre. The associated pCO_2 would imply an outgassing if those waters in the density layer in the subtropics were brought to the sea surface (Figures 6d and S4c). However, there is a downstream dilution of the DIC signal so that when the waters outcrop in the subpolar gyre, the associated pCO_2 is lower (Figure S4c) and implies an ocean uptake of CO_2 when compared for the atmospheric pCO_2 for 2024.

This ocean uptake of CO_2 is affected by the disequilibrium of the anthropogenic carbon. We illustrate this effect by comparing the actual pCO_2 and a hypothetical value if the anthropogenic pCO_2 was saturated in the layer (Figure 4d,e), and their comparison reveals that the undersaturation of anthropogenic carbon acts to lower pCO_2 from -125 to -25 μatm .

New text: L118-142

A biogeochemical stream is proposed following a sub-surface pathway from the subtropics to the subpolar latitudes (consistent with Fig. 5c). A reconstruction of that biogeochemical pathway from climatology reveals surface-depth property contrasts in the subtropics that weaken over the subpolar latitudes (Fig. 6a). Sub-surface waters have high concentrations of DIC in the subtropics, particularly along neutral densities 26.5 to 27.5 kg m⁻³, and there is a downstream decrease in their concentration as density layers shoal and outcrop in the subpolar gyre (Figs. 6a and S4a). Sub-surface waters likewise change from having low concentrations of anthropogenic carbon in the subtropics to progressively higher concentrations in the subpolar gyre (Figs. 6b and S4b). Accordingly there is a high potential for uptake of anthropogenic carbon in the subtropics to lower potential for uptake in the subpolar gyre, particularly along the neutral density layer 27 to 27.5 kg m⁻³, as those waters take up more anthropogenic carbon at the sea surface (Figs. 6c and S4b). This transport of DIC affects the difference in the partial pressure of the dissolved CO₂ and the atmospheric partial pressure

This transport of DIC affects the difference in the partial pressure of the dissolved CO₂ and the atmospheric partial pressure pCO₂, represented by ΔpCO₂, which is evaluated for year 2002 to be consistent with the timing of the ocean data. There are high values of ΔpCO₂ along neutral densities 26.5 to 27.5 kg m⁻³ in the subtropics (typically 80 to 150 μatmos for year 2002, Methods), suggesting that if those waters were locally brought to the surface, then at that time there would be an outgassing of CO₂ (Figs. 6d and S4c). There is a downstream dilution of the DIC signal along the same density layer that leads to the subpolar ΔpCO₂ being much smaller (typically ranging from 17 to 44 μatmos when evaluated for year 2002). Our reconstruction of the ΔpCO₂ signal is sensitive to the timing given the ongoing rise in atmospheric CO₂ (with an atmospheric increase of 50 μatmos between year 2002 and 2024). The subpolar values of ΔpCO₂ change from being positive for year 2002 to negative for year 2024 (typically ranging from -6 to -33 μatmos for year 2024) (Fig. S4c). Hence, there is an implied subpolar ocean uptake of CO₂ from the atmosphere for the present day.

The undersaturation of anthropogenic carbon affects this ΔpCO₂ signal and resulting air-sea flux. For example, if there is a saturation of anthropogenic carbon, then there would be a higher DIC value and associated ΔpCO₂ (Fig. 6e). The effect of the undersaturation of anthropogenic carbon on the ΔpCO₂ is represented here by the difference in the actual ΔpCO₂ and this hypothetical value when there is saturation: this effect varies from typically -125 μatmos in the subtropics to -25 μatmos in the subpolar latitudes using an atmospheric pCO₂ for year 2002 (Fig. 6f). Hence, the undersaturation of anthropogenic carbon is significant in determining the sign of the ΔpCO₂ signal over the subpolar latitudes (Figs. 6d and S4c).

New text in Methods

L219 to 224

The estimates of the biogeochemical properties following a subtropical-subpolar pathway (Fig. 6a-c) are evaluated using DIC and anthropogenic carbon reconstructions from gridded GLODAPv2 climatology60. The estimates of ΔpCO₂ (Fig. 6d-f) are from a difference between (i) ocean pCO₂ along a neutral density layer, 27.0 to 27.5 kg m⁻³, consistent with the DIC distribution and (ii) atmospheric pCO₂ of 373.1 μatmos for year 2002 and 422.8 μatmos for year 2024; the values reported in the text are based upon a 8° running latitudinal mean of the ocean pCO₂ values in Fig. S4c.

- While solubility pump is mentioned, biological pump is not discussed in the manuscript at all. The manuscript showed the importance of lateral transport and shoaling of high carbon uptake potential toward the downstream of the Gulf Stream. Yet, it is not clear to me how these two pumps work in the atmospheric CO₂ uptake.

We agree that we need to strengthen the context of the study and be more explicit as to how both the solubility pump and the biological pump operate. The transport of nutrients does affect the downstream biological activity, which contributes to the natural carbon uptake.

We have included new panels in Figure 1c-f to show the summer-winter nutrient utilisation and how the atmospheric CO₂ flux is split into thermal and biophysical contributions.

New text L56-58

While the effect of the Gulf Stream on the climate system has had extensive investigation, there has been less attention on its effect on biogeochemical cycles, such as on basin-scale patterns in seasonal nutrient utilisation (Fig. 1c) and annual air-sea CO₂ exchange over the North Atlantic (Fig. 1d), which includes both thermal and biological contributions (Fig. 1e,f).

Figure 1. Gulf Stream environment. Surface fields for the North Atlantic: (a) dynamic height (m) for year 2022 (from Operational Mercator global ocean analysis and forecast system¹⁶ at 1/12° horizontal resolution) with red arrows providing a schematic view of western boundary flows and blow arrow the North Atlantic Current; (b) net air to sea heat flux for winter (negative is ocean heat loss) at 1/4° resolution¹⁷ (W m^{-2}); (c) winter to summer nitrate contrast¹⁸ at the surface ($\mu\text{mol kg}^{-1}$); (d) annual sea-to-air CO₂ flux at 1° resolution ($\text{mol m}^{-2}\text{yr}^{-1}$) (positive is out of the ocean, negative is into the ocean)¹⁹, which is separated into (e) thermal and (f) biophysical components.

- In the Introduction it is indicated that Gulf Stream contribution to subpolar biological production is not well-recognized ‘The Gulf Stream is a major part of the upper limb of the Atlantic Meridional Overturning Circulation, [...] its contribution to mitigating human-driven atmospheric carbon increases and sustaining subpolar biological productivity is not’. However, the part of sustainability of biological production to higher latitudes is relatively well-known by in bibliographies in the manuscript such as references [31, 33]. Perhaps, it would be better to review the previous studies (some of the studies are actually by the authors) more precisely.

We have finessed the text to say

L29-31 The Gulf Stream is a major part of the upper limb of the Atlantic Meridional Overturning Circulation, and while the role of the Gulf Stream in the northward transport of heat is widely recognised, its contribution to mitigating human-driven atmospheric carbon increases is not.

We provide a review of the contributions of the Gulf Stream to nutrient transfer in L61-80.

- The following part in the Introduction sounds it doesn’t belong here since this paragraph is about previous studies that are not directly related to the present study.

“There is an ongoing debate about the importance of this ocean heat release to the atmosphere in determining the relatively mild winter climate of western Europe^{23–26}. Contrary to widespread perception, there is a view that the Gulf Stream is unimportant in explaining how western Europe is warmer in winter than an equivalent latitude on the eastern side of North America²³. Instead, the

warming of western Europe is explained by the atmospheric wave pattern set up by orography and seasonal heat release from the ocean to the atmosphere during winter. However, this view is challenged by an argument that seasonal heat release from the local ocean is insufficient to explain the warming and needs to be augmented by the heat supply from ocean transport involving the Gulf Stream²⁴. Alternatively, there is an intermediate view that warming from the ocean generates an atmospheric wave pattern leading to cold air on the eastern seaboard of North America and warmer air over western Europe and, thus, explains the winter temperature contrast between western and eastern continental boundaries²⁷".

We agree with the concern raised and have edited down this aspect.

L52-L55

This heat transfer is important in warming the atmosphere with typically half of the ocean heat carried by the meridional overturning circulation at 25°N being taken up by the atmosphere by 50°N^{14, 25}. This ocean transport and release of heat to the atmosphere²⁶, augments the seasonal heat release from the ocean to the atmosphere²⁷, and together contributes to the mild winter climate of western Europe.

Minor Comments

- It is pointed out during winter season (reference 28) the potential of carbon uptake by the ocean (Gulf Stream), what about other seasons? Is there (un)published information about similar potential in other time of the year? Although this is mainly for heat fluxes.

There will be a strong seasonality in terms air-sea heat and CO₂ fluxes, and biological activity.

What though is important is how the winter mixed layer properties are determined and the effect on the annual uptake of CO₂.

We now include in figure 1, panels of the annual air-sea CO₂ flux and its thermal and biological contributions.

- Figure 4 is a little confusing, as (a) belongs to 2019-2021, (b) from 1998 occupation, (c) corresponds similar to b I guess? And as for (e-f), values of b&c are combined with velocity values from 2000-2020 from (d)? Although in the manuscript it is clear the relation of the current and the potential outcropping of subsurface waters, including nitrate and carbon uptake potential, the period where the values are averaged are a little confusing. Also, the numbers and letters in Figure 4 are quite small (need to zoom in a lot to carefully read the values in isopycnals and axis).

We have modified this figure to make the velocity and property fields internally consistent.

L207-213. The nutrient and carbon observations across Florida Straits are from a series of hydrodynamical cruises along 27°N in 2012. The potential of waters to absorb additional anthropogenic carbon from the atmosphere, referred to as carbon uptake potential ($\mu\text{mol kg}^{-1}$), is estimated as the difference between dissolved inorganic carbon concentration calculated using in situ alkalinity, salinity, temperature, silicate and phosphate data and $p\text{CO}_2$ of 280 μatm (for the time of the preindustrial) or 373.1 μatm (for the time of the data collected in 2002) using CO2SYS5259 and the estimated in situ anthropogenic carbon concentration in 2002. Property transport (in kmol s^{-1}) were calculated by combining the property values for 2012^{38, 39} with average water velocities (in m s^{-1}) for the same time period based upon LADCP surveys of Florida Straits.

Figure 4. Biogeochemical Stream in the Florida Straits at 27°N: (a) salinity (psu) (with white contours for 12°C and 24°C); (b) nitrate concentration ($\mu\text{mol kg}^{-1}$); and (c) preformed nitrate ($\mu\text{mol kg}^{-1}$); (d) meridional velocity (m s^{-1}); (e) meridional nitrate transport (kmol s^{-1}); (f) meridional preformed nitrate transport (kmol s^{-1}); (g) dissolved inorganic carbon (DIC) ($\mu\text{mol kg}^{-1}$); (h) anthropogenic carbon ($\mu\text{mol kg}^{-1}$); (i) capacity of waters to absorb additional anthropogenic carbon (anthropogenic carbon uptake potential, $\mu\text{mol kg}^{-1}$); (j) northward DIC transport (kmol s^{-1}); (k) northward anthropogenic carbon transport (kmol s^{-1}); and (l) northward anthropogenic carbon uptake potential transport (kmol s^{-1}). The transports are calculated by combining property fields with average water velocities from a series of hydrodynamical cruises in 2012^{38,39} and the plots include neutral density contours 26.5 to 27.5 kg m^{-3} .

- Sadly, the link to the video of Additional information is not available anymore.

The video is on YouTube

<https://www.youtube.com/watch?v=dyVdmvvAjkc>

Or try searching via Google for Gulf Stream and carbon cycle.

- In Method: "... 1998 property field with average water velocities"

Updated to L212-213

Property transport (in kmol s^{-1}) were calculated by combining the property values for 2012^{38, 39} with average water velocities (in m s^{-1}) for the same time period based upon LADCP surveys of Florida Straits.

Using references

38. Wanninkhof, R. et al. Ocean acidification along the Gulf Coast and East Coast of the USA. *Cont. Shelf Res.* 98, 54–71312 (2015).

39. Zhang, J.-Z., Baringer, M. O., Fischer, C. J. & Hooper V, J. A. An estimate of diapycnal nutrient fluxes to the euphotic zone in the Florida Straits. *Sci. reports* 7, 16098 (2017).

This needs "." at the end.

Corrected

Also, please make the explanation better to be easy to understand for non-specialist, why taking this difference in DIC means anthropogenic carbon uptake.

We now include a supplementary figure S3e showing the profile of DIC (panel e, blue line) and anthropogenic carbon (panel i, blue line), and their difference then defines the preindustrial DIC (panel e, red line).

Figure S3. Profiles of biogeochemical profiles and transports in the Florida Straits at 27°N (as in Fig. 4): (a) nitrate (blue line) partitioning into preformed (orange line) and remineralised (yellow line) nutrients ($\mu\text{mol kg}^{-1}$); (b) nitrate transport integrated from the surface versus depth (kmol s^{-1}); (c) nitrate in density coordinates ($\mu\text{mol kg}^{-1}$); and (d) nitrate transport integrated from the surface versus neutral density (kmol s^{-1}); (e) DIC (blue line) and pre-industrial DIC (orange line) ($\mu\text{mol kg}^{-1}$); (f) DIC transport integrated from the surface versus depth (kmol s^{-1}); (g) DIC versus density ($\mu\text{mol kg}^{-1}$) and (h) associated transport integrated from the surface versus neutral density (kmol s^{-1}); (i) anthropogenic carbon (blue line) together with saturated anthropogenic carbon (orange line) that is the maximum possible anthropogenic carbon that can be held and potential to uptake additional anthropogenic carbon (yellow line) ($\mu\text{mol kg}^{-1}$); (j) anthropogenic carbon transport integrated from the surface versus depth (kmol s^{-1}); (k) anthropogenic carbon versus density and (l) associated transport integrated from the surface versus neutral density (kmol s^{-1}).

- In Method: “Every month 2% of the particles are randomly selected and their position reset to Florida Strait”.

Why randomly selected 2% of the particles only are relocated to the Florida Strait? Please elaborate on this.

Relocating 2% (or adding 2% of new particles at Florida Strait) every month suffices to establish a continuous source of particles at Florida Strait, and thus characterize the climatological flow of materials through the region.

L218-219

To ensure a continual source of particles, every month 2% of the particles are randomly selected and their position reset to Florida Strait.

Also, is vertical mixing considered in the particle tracking? Please justify it if it is not.

Convective, diapycnal, and isopycnal mixing could all in principle be represented via stochastic terms, but this is not attempted in the publications that we cite for comparison (e.g., Burkholder and Lozier 2011, Foukal and Lozier 2016, Rousselet et al 2021) or in our calculation. This choice seems justified in light of the state of the oceanographic literature, and notably given that there is no consensus on how to adequately implement stochastic terms in Lagrangian ocean frameworks. Here we instead present complementary Eulerian model diagnostics (i.e., the adjoint model) that do parameterize all major mixing processes in standard Eulerian fashion.

Thank you for your support for the article. We feel addressing your concerns has strengthened the manuscript.

We thank the editor for provisionally accepting the manuscript and the referees for positive comments. Only 1 referee had remaining concerns and we have attempted to address that concern in full.

REVIEWERS' COMMENTS:

Reviewer #1 (Remarks to the Author):

The authors still do not understand the difference between CO₂ and the anthropogenic CO₂, which is only about 3% of the total CO₂. The two primary "carbon pumps", namely the biological pump and the physical pump, move carbon regardless of whether it is natural or anthropogenic. Much of the description of the anthropogenic carbon is wrong. For instance, they compare the waters' anthropogenic carbon concentration with the modern atmosphere. This comparison has nothing to do with whether the water takes up carbon, anthropogenic or not.

We have included more material in the main manuscript to provide the additional context needed:

Figure 1 expanded to include net surface heat fluxes and winter surface nitrate to provide more context for the thermal and biophysical drivers of ocean carbon uptake;

Figure 4 includes profiles of nitrate, dissolved inorganic carbon and anthropogenic carbon;

Figure 6 expanded to include nitrate and partial pressure of carbon dioxide CO₂ together with scatterplots of dissolved inorganic carbon DIC, anthropogenic carbon and pCO₂.

Supplementary figures reduced from 4 figures to 3 figures (due to moving information into main manuscript).

A new Figure S3 includes complete set of biogeochemical sections across the Florida Straits.

In terms of text, we have added a new paragraph L128-L137, P9 that addressed the point of the referee:

The air-sea flux of CO₂ is locally defined by the difference in the partial pressure of the dissolved CO₂ in the surface mixed layer and the atmospheric partial pressure, pCO₂, represented by $\Delta p\text{CO}_2$ (Methods). The dissolved CO₂ signal in the mixed layer is affected by the advection of DIC and accompanying temperature and nutrients along density layers and eventual outcrop of these waters (Fig. 6c and e, red bars). Over the subpolar latitudes, the $\Delta p\text{CO}_2$ signal is positive along the neutral densities 27.0 to 27.5 kg m⁻³ and represents waters that are just over saturated for year 2002 (Fig. 6e, red bars and grey dashed line). Hence, by itself, this injection of DIC along the density surface into the mixed layer elevates surface pCO₂ values, and suggests subpolar CO₂ outgassing for 2002. However, there are additional thermal and biological mechanisms that subsequently affect the surface pCO₂ values: in particular, surface heat loss together with biological utilisation of the nutrients (predominantly preformed) additionally delivered to the mixed layer (Fig. 1d-f and b), each lead to a lowering of surface pCO₂ values, generating a negative $\Delta p\text{CO}_2$. A net annual ocean uptake of CO₂ from the atmosphere from both thermal and biophysical drivers is thus derived 44, 45 (Fig. 1g-i).

We also then make our assertion with supporting evidence that there undersaturation of anthropogenic carbon plays a contributing role (L138-148, P9):

The total carbon content of these waters also shapes the magnitude of the uptake signal. As the biogeochemical stream waters have been separated from the atmosphere over long timescales, these waters are undersaturated with respect to a modern atmosphere. Sub-surface waters change from having low concentrations of anthropogenic carbon in the subtropics to progressively higher concentrations in the subpolar gyre along neutral densities 27.0 to 27.5 kg m⁻³ (Fig. 6f, orange line and g). This undersaturation (Fig. 6f, red bars and h) gives the waters a greater capacity to absorb additional CO₂ from the atmosphere when they outcrop into the mixed layer (and where thermal and biological processes drive pCO₂ levels down in the surface ocean). The effect of this anthropogenic carbon undersaturation can be represented by the difference between the actual pCO₂ and a hypothetical pCO₂ the waters would have if saturated (Fig. 6e, red and black bars respectively). The undersaturation of anthropogenic carbon leads to a more negative $\Delta p\text{CO}_2$ that varies from typically -125 μatm in the subtropics to -25 μatm in the subpolar latitudes (Fig. 6i). Hence, the undersaturation of anthropogenic carbon in the waters outcropping in the subpolar gyre enhances the subpolar ocean uptake of atmospheric CO₂.

A second issue is whether upwelling leads to uptaking or degassing of CO₂. I used the Kuroshio example to show that upwelling may or may not lead to degassing. The authors are correct that their study area is the Gulf Stream, but no similar study appears to exist in the Gulf Stream. Note that uptaking or degassing of CO₂ has nothing to do with whether the CO₂ is anthropogenic.

There is no disagreement here.

Reviewer #2 (Remarks to the Author):

Second Review of “The biogeochemical stream of the Gulf Stream”, formerly entitled “Biogeochemical streams supply nutrients and depleted anthropogenic carbon over the North Atlantic”, by R. G. Williams et al., submitted to Communications Earth & Environment

This is a second review of the manuscript entitled “The biogeochemical stream of the Gulf Stream”, formerly entitled “Biogeochemical streams supply nutrients and depleted anthropogenic carbon over the North Atlantic”, by R. G. Williams et al. As exposed in my original review, I believe this is a timely manuscript that looks at the relevance of biogeochemical processes in the Gulf Stream. In this new version, the authors have done a great job, I would like to thank them for their efforts and congratulate them for the outcome. In particular, they have addressed a main issue raised in my earlier review, namely they have provided a much better description of the along-stream changes in both total dissolved inorganic carbon (DIC) and its anthropogenic fraction, which is necessary to better elucidate what will be the ocean carbon uptake caused by an increase in atmospheric CO₂.

In their response to my earlier review, the authors have also included several comments on which I may disagree, but these do not affect substantially this new manuscript version so I will only refer to one specific issue. Palter and Lozier (2008) used an oxygen-corrected phosphate tracer to show that intermediate layers (potential density in the range 26.0 to 27.0) have a >80% subantarctic contribution, but this is not the same as saying that 80% of the nutrient load in these layers is of subantarctic origin (line 76 of the manuscript). Figure 3c, in my opinion, does not evidence this western-boundary transport continuity from the tropics to the subtropical North Atlantic.

Agreed. Our comments referred to the source of preformed nutrients, while the referee was emphasising the nutrient load that includes the input of regenerated nutrients.

We have included extra panels to show the profiles and sections of preformed and regenerated nutrients in Figure 4 and Figures S2 and S3.

I have only a couple of additional minor suggestions for the authors' consideration:

1) In my opinion, repeating the word “stream” in a seven-words title sounds rare. I prefer not to make any suggestion but I leave it to the authors, and the editor, for consideration.

Dealt with through new title.

2) In line 90 it is said that about half of the nutrient concentrations are preformed and half remineralized, referring to figure 4. However, the results for nitrate flux in Figures 4e,f suggest that nitrate preformed flux is relatively small as compared to the total.

We show the nutrient transports in Figure S2b,d, the preformed nitrate transport is typically 40% of the total nutrient transport.

I would like to congratulate the authors for a nice piece of research, I'm happy to recommend publication in the journal.

Josep L. Pelegrí

Thank you.

Reviewer #3 (Remarks to the Author):

Overall I am quite satisfied with the authors' responses to the issues raised in the last review. The additional review and discussion helps clarify the manuscript's main points.

While it may be beyond the main scope of this work, I would note in regard to the low light level point, that many field programs are increasingly finding photosynthesis occurring at extremely low light levels and this may play a role here as in: <https://www.nature.com/articles/s41467-024-51636-8>. But it is a good point that if these waters stay below the summer mixed layer the effect will be delayed into the winter.

The notes and increased description of the sensitivity analysis are helpful. And while the adjoint model of course effectively represents many perturbations in the system, there is an underlying assumption that the model architecture (and the parameters and parameterizations of that model) can effectively represent this system, which may or may not be true and this is why I suggested the more cautious phrasing. But regardless, I accept the authors' preference for their current phrasing and believe it is reasonable.

Thank you.

Some figures are still fairly low resolution (e.g. the x and y axis labels and contour labels on Figure 4 are very low resolution even though the subplot titles are high res) but I imagine this can be quickly fixed in the final proofing process.

Figures 1,2, 3, 4 and 6 have been replotted.

I look forward both to seeing this work published and to reading your future reports on the field program you describe.

Reviewer #4 (Remarks to the Author):

The authors have revised all questions

Thank you.

Review of “Biogeochemical streams supply nutrients and depleted anthropogenic carbon over the North Atlantic” by R. G. Williams et al., submitted to Communications Earth & Environment

The manuscript entitled “Biogeochemical streams supply nutrients and depleted anthropogenic carbon over the North Atlantic”, by R. G. Williams et al., addresses the relevance of the Gulf Stream as a major subsurface conduit for the transport of biogeochemical properties from the subtropical to the subpolar North Atlantic, in what constitutes the last stretch of the upper or returning limb of the Atlantic Meridional Overturning Circulation (AMOC). The manuscript reviews ideas developed during the last decades on the key role of the Gulf Stream exporting heat and (inorganic) nutrients from the tropics and subtropics into the subpolar North Atlantic, and further explores more recent developments on the associated transport of dissolved inorganic carbon (DIC) in relatively old waters, depleted in anthropogenic carbon, and their potential impact on climate. (Hereafter, when saying “nutrients” I will always refer to “inorganic nutrients”.)

In my view the manuscript has two main relevant contributions:

- 1) It reviews historical concepts, and revisits them with several new plots, in a comprehensive way, hence providing a joint view on how the Gulf Stream transports three properties that are key for climate – heat, nutrients and DIC, specifically DIC anomalies of anthropogenic origin – into the subpolar North Atlantic.
- 2) It extends the relatively recent work by S. Ridge and G. McKinley (Advective controls on the North Atlantic anthropogenic carbon sink. *Glob. Biogeochem. Cycles* 34, e2019GB006457, 2020) on the supply of subsurface waters depleted in anthropogenic carbon to the subpolar North Atlantic, exploring the extra-subtropical sources.

Placing these ideas together has substantial merit and would likely grant the publishing of this manuscript. However, in my view, there are a number of significant aspects related to the above main contributions that ought to be completed and/or clarified before the manuscript is published. I will try to explain them next, followed with an enumeration of additional minor issues that I believe should also be considered.

Historical concepts on heat and nutrient transport

The subsection on **Gulf Stream and heat transfer** begins referring to the Gulf Stream heat transport by citing relatively new references. Here I miss the mentioning of early seminal work on northward heat transport by Carl Wunsch and collaborators (e.g. Wunsch and Grant, 1982; Wunsch, 1984; Rintoul and Wunsch, 1991).

In the third line of the next subsection, on **Gulf Stream and downstream nutrient transfer**, the authors state “While there is a clear sea surface temperature signal, there is little contrast in surface nutrient concentrations (Fig. 2b).” Figure 2 does not show the temperature field, so there is no way to know if there is a temperature contrast between the Gulf Stream and the gyre interior (I imagine this is what the authors mean), and it is difficult to appreciate if the surface nutrient contrast is significant. In my opinion, the important point here is that there is little along-isopycnal contrast in the concentration of nutrients, this should be clearly stated.

The authors state that the “The nutrient stream provides a nutrient communication pathway over the basin” and cite Pelegrí and Csanady (1991) and Pelegrí et al. (1996). I think this should be clarified, the basin-wide nutrient communication responds to the subtropical along-isopycnal anticyclonic basin-wide recirculation and the associated progressive remineralization

of the organic matter that is subducted at higher latitudes. The outcome is the “nutrient bearing stratum (NBS)” and, in particular, the low isopycnal contrast in nutrient concentration between the core of the nutrient stream and the interior subtropical waters. The work by Pelegrí and Csanady (1991) and Pelegrí et al. (1996) focused on how this along-isopycnal basin-wide clockwise increase in nutrients is channeled northward via the Gulf Stream as a subsurface along-isopycnal stream of nutrients – which was progressively depleted as waters reached the sea surface – and pioneered the calculations on the intensity of the nutrient stream, the associated nutrient transport and its northward penetration and irrigation of the subpolar region. I believe this is also a good opportunity for the authors to briefly refer to the original work behind the NBS, in particular work by Jorge Sarmiento and collaborators in the 1980s, as well as the review by Gabe Csanady (Physical basis of coastal productivity: The SEEP and MASAR experiments, EOS, 71, 1060-1065, 1990) where the NBS term was first coined.

Subtropical versus remote water, nutrient and carbon sources

The source of the Gulf Stream waters with high concentration of nutrients (and low concentration of anthropogenic-carbon depleted carbon, see below) is, or should be, a central issue of this manuscript. The authors provide several pieces of information but I feel they fail to close the story in a convincing manner, or at least to clearly point at what we still do not know about this issue.

In the section on **Gulf Stream and downstream nutrient transfer**, the authors state that “The nutrients carried in the nutrient stream mainly originate from outside the subtropics, and are supplied from mode and intermediate waters originating from the Southern Ocean”, citing Sarmiento et al. (2004) and Palter and Lozier (2008). Indeed, these authors proved that subantarctic mode waters are the main remote source of waters arriving to the Gulf Stream waters through the Florida Straits but this is not the same to saying that the nutrient load of the Gulf Stream is dominated by nutrients preformed in the subantarctic ocean. The argument somehow parallels heat transport, the returning limb of the AMOC comes from the subantarctic region but we know that most heat transport by the Gulf Stream does not come from that far.

Indeed, several arguments show that most of the nutrient load in the nutrient stream likely has its source on the subtropics. First, the actual upper-limb contribution of the AMOC is no more than 20 Sv so necessarily a substantial fraction of the northward flow through the Florida Straits (about 35 Sv) and east of Bahamas (less than 5 Sv) is subtropical water recirculating through the Sargasso Sea. Second, between the Florida Strait and 36°N the Gulf Stream (down to 2000 m) roughly doubles its water transport but triples its nutrient transport (Pelegrí and Csanady, 1991); this additional nutrient-rich water source is essentially nutrient-rich water recirculating in the subtropical gyre. Third, the along-isopycnal continuity of nitrate concentration (Fig. 2b) and the subtropical pathways in the submitted manuscript (Figure 5) also support the idea that most of the nutrient stream has its source on the subtropics, with the highest nutrient concentrations corresponding to water recirculating through the long Sargasso-Sea subtropical pathway. Fourth, the works by Williams, McDonagh, Roussenov et al. (Nutrient streams in the North Atlantic: Advective pathways of inorganic and dissolved organic nutrients. *Global Biogeochemical Cycles*, 25, GB4008, 2011) and Holzer and Primeau (Global teleconnections in the oceanic phosphorous cycle: Patterns, paths, and timescales. *Journal of Geophysical Research Oceans*, 118, 1775-1796, 2013) show that the load of subantarctic nutrients incorporated into the Gulf Stream is relatively small.

Fifth, and final, Figure 3 of the manuscript is interpreted by the authors as implying that “nutrients carried in the nutrient stream mainly originate from outside the subtropics, and are

supplied from mode and intermediate waters originating from the Southern Ocean". The **Methods** section in the manuscript does not explain how this is done but I understand that the figure shows transports in 0.23° bins in the isopycnal layer between 26.8 and 27.5 kg/m^3 . If this is the case, I cannot agree that the figure implies a major extra-subtropical source. The figure is difficult to interpret (there are large complex gyres, and I wonder if there is a white color in the color bar?) but I would say that the northern transport at (25°N , 80°W) and (25°N , 75°W) is very intense but also very narrow, while the western transport at 70°W is also intense but very wide, extending between 23°N and 27°N , largely feeding from the subtropical cycle. To actually see these contributions a plausible strategy would be to calculate and plot the zonally integrated values of meridional nitrate transport.

I would like to also suggest the authors to look at the review article by Pelegrí et al. (The Gulf Nutrient Stream. In Kuroshio Current: Physical, Biogeochemical and Ecosystem Dynamics, Eds. T. Nagai, H. Saito, K. Suzuki, M. Takahashi. AGU-Wiley, GMS 243, 23-50, ISBN 9781119428343, 2019), as it is not cited in the manuscript despite it contains an exhaustive revision of many of the above ideas.

A source of subsurface waters depleted in anthropogenic carbon

A good understanding of the partition between subtropical- and remote-source contributions is crucial when we look at the Gulf Stream transport of subsurface waters depleted in anthropogenic carbon (C_{ant}). In my opinion, here it is very important to consider the time scales of the different pathways, as these define their "age", i.e. their C_{ant} at origin and hence their uptake capacity. Waters that are part of the upper (returning) limb of the AMOC and were subducted in the subantarctic region will take many years to reach the Gulf Stream (as they may recirculate in the subtropical and tropical oceans several times), while waters subducting in the North Atlantic will take much less to reach the Gulf Stream.

For simplicity, suppose these times are 50 and 10 years, respectively, so to assess today's absorbing capacity of these different waters the relevant figures would be the CO_2 atmospheric concentrations in 1974, 2014 and 2024, which respectively are 330, 399 and 425 ppm. The actual absorbing capacity of the subsurface Gulf Stream waters reaching the subpolar ocean will hence depend on a proper assessment of the travel times and relative contributions of the different water masses.

In the section **Sensitivity of subpolar carbon to upstream waters** the manuscript explores the extension of the upstream region that influences the carbon content in the subpolar North Atlantic at 4 and 8 years (or something alike, I would appreciate a better description of what the authors mean by "sensitivity", I cannot either see a proper explanation in the **Methods**). These are relatively short times (it would be nice to see the results for longer times) but nevertheless we appreciate an upstream influence into both the Caribbean and the northwestern portion of the gyre, in what would constitute a smaller recirculation region of the subtropical gyre (see Figure 9 in Pelegrí et al., 2019). To me this suggests that there is a subtropical source (as proposed by Ridge and McKinley, 2020) of relatively young waters, so with relatively little uptake capacity, as well as a remote source of much older waters, largely corresponding to the upper limb of the AMOC.

Additional minor issues

- In the **Introduction**, the authors refer to the Gulf Stream as a "biogeochemical stream" that "provides a sub-surface redistribution of nutrients and depleted anthropogenic carbon... which extends over the North Atlantic". Two issues appear here. First, to my knowledge, this is the

first time the “biogeochemical stream” terminology is used, so it should be stated clearly. Second, the term “biogeochemical stream” is, in my opinion, incomplete as “biogeochemical” is an adjective (such as “geochemical”, Oxford Dictionary), so “biogeochemical streams” refer to streams that have some “biogeochemical” characteristics but without stating which ones. This contrasts with “nutrient streams”, as “nutrient” is both an adjective and a noun (Oxford Dictionary), so “nutrient stream” becomes a compound noun, a stream that has or carries nutrients.

- The **Introduction** ends up saying “future possible changes in carbon uptake from the atmosphere are illustrated for the North Atlantic that are separated into the ocean carbon response and feedback due to rising atmospheric CO₂ plus the ocean carbon feedback due to climate change”. The two concepts are intimately related so, to avoid confusion, I suggest that you clarify what you mean by “ocean carbon feedback due to climate change”.

- Line 5 of subsection **Gulf Stream and heat transfer** reads “there is a maximum in heat loss to the atmosphere from turbulent heat fluxes”; I believe this should read “there is a maximum in heat loss to the atmosphere from air-to-sea turbulent heat fluxes”.

- I believe that the last two sentences of this same paragraph (“The Gulf Stream makes a crucial... taken up by the atmosphere by 50°N”) contain information related to the beginning of this same paragraph, so they appear somewhat misplaced.

- The second paragraph of this subsection (starting with “There is an ongoing debate...”) is, in my opinion, confusing. I would suggest rewriting and clarifying the ideas in this paragraph.

- In section **Connectivity of the Gulf Stream and the subpolar gyre** the manuscript refers to particles released in the Florida Strait at several depth ranges. I think that, more than a depth range, the relevant issue is the isopycnal range, as the water penetration of the Gulf Stream extension into the subpolar gyre is largely set by the outcropping of isopycnals, and in the Florida Straits the isopycnals are very tilted with e.g. the 27.0 isopycnal ranging between 100 and 550 m. In my opinion, this creates uncertainties in the interpretation of the results.

- In this same section, at the end of page 4 and beginning of page 5, the manuscript reads “Sub-surface waters have high nitrate concentrations in the subtropics and there is a downstream surface increase in the nitrate as density layers shoal and outcrop in the subpolar gyre (Fig. 6a).” I believe this is not correct: the extension of the nutrient stream into the subpolar gyre is indeed accompanied by a wedge of nutrients, what Pelegrí et al. (2006) called an irrigation of nutrients, but the nutrient concentration does decrease along this pathway.

- Figure 6: The distribution of properties along the selected pathway is difficult to appreciate.

- In the section **Sensitivity of subpolar carbon to upstream waters** the manuscript mentions an “adjoint model” and the reader is referred to the Methods section, but this section does not explain why calling it “adjoint”. I had to go to the original reference and spend some time there to get an idea of what is meant by adjoint, I would suggest to include a short explanation to facilitate the reading of the manuscript.

- A few lines later the manuscript talks about “carbon carried in a potential density layer of $\sigma_\theta = 27.5 \text{ kg/m}^3$.” This is an inconsistent sentence: a layer is, by definition, a range, and a density surface (such as the 27.5 kg/m^3) has infinitesimal thickness and hence cannot carry properties. Please consider rewriting.

- Figure 7, caption: the variable J is introduced but not defined, the reader only finds out its definition after ending **Methods**.

- The second paragraph of **Methods** reads “carbon uptake potential ($\mu\text{mol kg}^{-1}$), is estimated as the difference between dissolved inorganic carbon concentration calculated using in situ alkalinity, salinity, temperature, silicate and phosphate data and pCO_2 of 280 μatm (preindustrial) or 369 μatm (1998 level) using CO2SYS52 and the estimated in situ anthropogenic carbon concentration in 1998.” This is confusing, please clarify why “280 μatm or 369 μatm ”, and why the anthropogenic rather than the total carbon concentration in 1998.

- It is possibly worth mentioning that the nutrient stream concept has also been widely used for the Kuroshio Current, see the book “Kuroshio Current: Physical, Biogeochemical and Ecosystem Dynamics”, edited by T. Nagai, H. Saito, K. Suzuki, M. Takahashi, AGU-Wiley, GMS 243, 23-50, ISBN 9781119428343, 2019. This is why, in their review article, Pelegrí et al. (2019) decided to use the nomenclature Gulf Nutrient Stream, which can hence be capitalized.

- Possibly the authors should also clarify that, except when indicated, nutrient streams always refer to streams of inorganic nutrients.

Recommendation

I admire and worthy very much the past and present work done by the authors on the origin, fate, intensity and biogeochemical processes in the nutrient streams, and I think this is a timely review that evidences the high relevance of biogeochemical processes in the North Atlantic nutrient stream, and further explores the role of this stream related to climate change. However, I think that the manuscript will greatly benefit by a revision that (a) solves the issues raised above, so as to facilitate its reading, and (b) clarifies what we know and what we yet do not know on the origin of those waters with high concentration of nutrients and low concentration of anthropogenic-carbon depleted carbon that feed the Gulf Nutrient Stream. Clarifying these different origins is crucial for establishing the CO_2 uptake capacity of the nutrient stream subsurface waters and its potential role controlling climate.

I'm happy to sign this review, Josep L. Pelegrí.